# Longitudinal body mass index and cancer risk: a cohort study of 2.6 million Catalan adults

Martina Recalde[1,2,3,7], Andrea Pistillo [2,7], Veronica Davila-Batista[1,4] ✉,
Michael Leitzmann[5], Isabelle Romieu[6], Vivian Viallon[1], Heinz Freisling [1,8] ✉ &
Talita Duarte-Salles [2,8] ✉

Single body mass index (BMI) measurements have been associated with increased risk of 13 cancers. Whether life course adiposity-related exposures are more relevant cancer risk factors than baseline BMI (ie, at start of follow-up for disease outcome) remains unclear. We conducted a cohort study from 2009 until 2018 with population-based electronic health records in Catalonia, Spain. We included 2,645,885 individuals aged ≥40 years and free of cancer in 2009. After 9 years of follow-up, 225,396 participants were diagnosed with cancer. This study shows that longer duration, greater degree, and younger age of onset of overweight and obesity during early adulthood are positively associated with risk of 18 cancers, including leukemia, non-Hodgkin lymphoma, and among never-smokers, head and neck, and bladder cancers which are not yet considered as obesity-related cancers in the literature. Our findings support public health strategies for cancer prevention focussing on preventing and reducing early overweight and obesity.

In 2016, 1.9 billion and 650 million adults were living with overweight and obesity, respectively[1]. Body mass index (BMI), the most common indicator to capture overweight (BMI ≥ 25 kg/m²) and obesity (BMI ≥ 30 kg/m²), has been convincingly associated with the risk of at least 13 cancer types[2]. However, previous studies have mostly focussed on single BMI measurements assessed at study baseline, which are measures of current BMI status. Whether overweight and obesity over the life course are more relevant risk factors for cancer remains unclear[3–5]. Capturing longitudinal BMI-derived exposures might better reflect the underlying biological mechanisms between long-term exposure to adiposity and cancer development. At an epidemiological level, this could translate into stronger associations between adiposity and obesity-related cancer risk and into adiposity being linked to a larger number of cancer types than currently recognized.

Few studies have investigated the association between longitudinal BMI-derived exposures and cancer risk[6–10]. These exposures included duration of years lived with a BMI ≥ 25 or ≥30 kg/m² and cumulative exposure (an indicator considering degree and duration of overweight/obesity) to a BMI ≥ 25 or ≥30 kg/m², which have been positively associated with risk of cancers of the colorectum, postmenopausal breast, endometrium, kidney, pancreas, and multiple myeloma[6–10]. Studies investigating age of onset of a BMI ≥ 25 or ≥30 kg/m² in relation to cancer risk are currently lacking. Yet, such knowledge could identify periods of age, when overweight/obesity are most relevant to cancer risk.

Prior studies have provided insights into the longitudinal BMI-derived exposures-cancer association but did not formally compare cancer risk estimates of longitudinal exposures to those of baseline

[1]International Agency for Research on Cancer (IARC-WHO), 25 avenue Tony Garnier, CS 90627, 69366 Lyon Cedex 07 Lyon, France. [2]Fundació Institut Universitari per a la recerca a l'Atenció Primària de Salut Jordi Gol i Gurina (IDIAPJGol), Barcelona, Spain. [3]Universitat Autònoma de Barcelona, Barcelona, Spain. [4]Spanish Consortium for Research on Epidemiology and Public Health (CIBERESP), Instituto de Salud Carlos III, 28029 Madrid, Spain. [5]Department of Epidemiology and Preventive Medicine, University of Regensburg, Regensburg, Germany. [6]Center for Research on Population Health, National Institute of Public Health, Mexico City, Mexico. [7]These authors contributed equally: Martina Recalde, Andrea Pistillo. [8]These authors jointly supervised this work: Heinz Freisling, Talita Duarte-Salles. ✉e-mail: vdavbat@gobiernodecanarias.org; FreislingH@iarc.fr; tduarte@idiapjgol.org

BMI. Other limitations involve excluding individuals without BMI information (increasing the risk of selection bias), having limited sample sizes that preclude the analysis of a wider range of cancers, or relying on self-reported and recalled weight and height, which could increase the likelihood of exposure misclassification. A study with BMI data measured by health professionals, capturing incident cancer cases from a large and representative population, and using advanced multiple imputation techniques to BMI for all eligible participants could help gain understanding of the adiposity–cancer association through a life-course perspective.

We investigated the association between duration of years lived with a BMI ≥ 25 and ≥30 kg/m², cumulative exposure to a BMI ≥ 25 and ≥30 kg/m², age of onset of a BMI ≥ 25 and ≥30 kg/m² during early adulthood (18–40 years) and BMI at baseline in relation to the risk of 26 cancer types.

In this work we show that longer duration and greater degree of overweight and obesity during early adulthood as well as younger age of onset of a high BMI are associated with a higher risk of 18 cancer types.

## Results

Of the 3,247,244 individuals who were eligible to enter the study, we excluded 172,800, 190,171, and 238,388 persons who had <1 year of history in SIDIAP, prior history of cancer, and <1 year of follow-up, respectively (Supplementary Fig. S1).

Among 2,645,885 participants followed up for a median time of 9 (interquartile range [IQR]: 8–9) years, 225,396 (9%) individuals were diagnosed with any of the 26 cancers of interest (Table 1). The median age of the participants was 56 (IQR: 47–68) years and 47% were males. Of the included participants, 2,081,840 (79%) had at least one BMI assessment in their electronic health records while 564,045 (21%) had none (Supplementary Table S1). Individuals without a BMI assessment were more likely to be men and younger compared to those with a BMI assessment. The former also had a higher proportion of non-Spanish, individuals living in the least deprived areas of Catalonia and people with no comorbidities, compared to the latter (Supplementary Table S1). The median (imputed) BMI at index date (baseline) was 28 (24–31) kg/m² for all participants (Table 1). The median duration of BMI ≥ 25 and ≥30 kg/m², respectively, were 12 (0–23) and 0 (0–4) years. The median cumulative exposure to BMI ≥ 25 and to BMI ≥ 30 m/kg² were 16 (0–74) cumulative overweight-years and 0 (0–2) cumulative obesity-years, respectively. Of all participants, 1,833,516 (69%) ever had a BMI ≥ 25 kg/m² (median age of onset of BMI ≥ 25 was 20 [IQR: 18–29] years), of which 801,612 (30% of all participants) ever had a BMI ≥ 30 kg/m² (median age of onset of BMI ≥ 30 was 29 [21–35] years).

**Table 1 | Baseline characteristics of the study population, overall and by having ever had a body mass index ≥ 25 or ≥ 30 kg/m², after multiple imputations**

| | Overall N (%) | Never overweight (BMI < 25 kg/m²)N (%)[1] | Ever with overweight (BMI ≥ 25 kg/m²) N (%)[1] | Ever with obesity (BMI ≥ 30 kg/m²) N (%)[1] |
|---|---|---|---|---|
| | 2,645,885 | 812,369 (30.7) | 1,833,516 (69.3) | 801,612 (30.3) |
| Follow-up time in years, median (IQR) | 9.0 (7.7, 9.0) | 9.0 (7.9, 9.0) | 9.0 (7.7, 9.0) | 9.0 (7.5, 9.0) |
| Duration of BMI ≥ 25 kg/m² in years, median (IQR)[2] | 12.0 (0.0, 23.0) | 0.0 (0.0, 0.0) | 20.0 (10.0, 23.0) | 23.0 (23.0, 23.0) |
| Duration of BMI ≥ 30 kg/m² in years, median (IQR) | 0.0 (0.0, 4.0) | 0.0 (0.0, 0.0) | 0.0 (0.0, 9.0) | 11.0 (5.0, 20.0) |
| Cumulative exposure to BMI ≥ 25 kg/m² in cumulative overweight-years, median (IQR)[2,3] | 16.4 (0.0, 73.7) | 0.0 (0.0, 0.0) | 45.9 (13.7, 103.3) | 113.5 (78.3, 163.3) |
| Cumulative exposure to BMI ≥ 30 kg/m² in cumulative obesity-years, median (IQR)[2,3] | 0.0 (0.0, 2.2) | 0.0 (0.0, 0.0) | 0.0 (0.0, 12.4) | 17.4 (4.2, 51.9) |
| Age of onset of BMI ≥ 25 kg/m² in years, median (IQR)[2,4] | 20.0 (18.0, 29.0) | - | 20.0 (18.0, 29.0) | 18.0 (18.0, 18.0) |
| Age of onset of BMI ≥ 30 kg/m² in years, median (IQR)[2,4] | 29.0 (21.0, 35.0) | - | 29.0 (21.0, 35.0) | 29.0 (21.0, 35.0) |
| BMI at index date in kg/m², median (IQR)[2,5] | 27.6 (24.2, 31.1) | 23.0 (20.7, 24.9) | 29.4 (26.8, 32.5) | 32.5 (30.5, 35.2) |
| Age in years, median (IQR) | 56.0 (47.0, 68.0) | 55.0 (46.0, 66.0) | 57.0 (47.0, 70.0) | 58.0 (48.0, 71.0) |
| Male sex, *n* (%) | 1,241,523 (46.9) | 362,147 (44.6) | 879,376 (48.0) | 358,172 (44.7) |
| **Nationality** | | | | |
| Spanish | 2,495,536 (94.3) | 766,176 (94.3) | 1,729,360 (94.3) | 756,163 (94.3) |
| Global North | 51,320 (1.9) | 17,049 (2.1) | 34,271 (1.9) | 14,834 (1.9) |
| Global South | 99,029 (3.7) | 29,145 (3.6) | 69,884 (3.8) | 30,616 (3.8) |
| **MEDEA deprivation index, *n* (%)[2]** | | | | |
| Quintile 1 (least deprived) | 472,049 (17.8) | 170,403 (21.0) | 301,646 (16.5) | 120,028 (15.0) |
| Quintile 5 (most deprived) | 361,665 (13.7) | 96,963 (11.9) | 264,702 (14.4) | 125,063 (15.6) |
| **Smoking status, *n* (%)[2]** | | | | |
| Never smoker | 1,663,154 (62.9) | 486,100 (59.8) | 1,177,054 (64.2) | 529,478 (66.1) |
| Former smoker | 390,711 (14.8) | 110,853 (13.6) | 279,858 (15.3) | 122,089 (15.2) |
| Current smoker | 592,020 (22.4) | 215,416 (26.5) | 376,604 (20.5) | 150,046 (18.7) |
| **Cancer outcomes, *n* (%)** | 225,396 (8.5) | 64,466 (7.9) | 160,930 (8.8) | 71,456 (8.9) |

Notes: (1) This categorization was done in the 5 datasets obtained after performing the multiple imputations. For visualization purposes and in order for the categorical variables to add up to 2,645,885 we divided the n for the categorical variables by 5. (2) The exposures of interest, the MEDEA deprivation index, smoking status, and alcohol intake were calculated using the multiple imputation approach, with 5 data sets created. For visualization purposes, we divided the n for the categorical variables by 5. (3) This indicator was calculated by adding the difference between the BMI measurements that were ≥25 (≥ 30, for obesity) kg/m² and 24.9 (29.9) kg/m² for every year lived with a BMI ≥ 25 and ≥30, respectively. (4) Age of onset of a BMI ≥ 25 (and ≥30) kg/m² is only available for individuals who ever had a BMI ≥ 25 (≥ 30) kg/m². (5) BMI assessment at the start of the time-to-event analysis (baseline BMI). A more complete description of the baseline characteristics of the study partipants can be found in Supplementary Table S6.

Abbreviations: *BMI* Body Mass Index, *IQR* Interquartile range, *MEDEA* Mortalidad en áreas pequeñas Españolas y Desigualdades Socioeconómicas y Ambientales *SIDIAP* Information System for Research in Primary Care.

Those who never had a BMI ≥ 25 kg/m² were more likely to live in the least deprived areas of Catalonia and to be current smokers than those who ever had a BMI ≥ 25 kg/m² (Table 1).

### Association between BMI-derived exposures and cancer risk

In fully adjusted models, longer duration of a BMI ≥ 25 (≥ 30) kg/m² was positively associated with the risk of 14 (12) cancers, higher cumulative exposure to a BMI ≥ 25 (≥ 30) kg/m² with 13 (11), age of onset of a BMI ≥ 25 (≥ 30) kg/m² with 11 (10), and BMI at index date with 10 cancers (Fig. 1, Supplementary Table S2 & S3). All exposures were positively associated with the risk of the following eight cancer types: corpus uteri (eg, HR, 95%CI per 10-year [1-SD] increment of duration of a BMI ≥ 25: 1.46, 1.42-1.51), kidney, gallbladder and biliary tract, breast postmenopausal, leukemia, multiple myeloma, colorectal, and liver (same eg: 1.04, 1.01-1.07) cancers. All exposures except age of onset of a BMI ≥ 25 and/or ≥30, were also positively associated with the risk of two cancers: thyroid (eg, HR, 95% CI per 70-cumulative overweight-year [1-SD] increment of cumulative exposure of a BMI ≥ 25: 1.08, 1.04-1.12), and brain and CNS (same eg: 1.06, 1.02-1.10). There were nuances in the shape of the relationship of some of the exposures with the risk of six of these cancers (p-value for non-linearity <0.05) (Figs. 2, 3, and 4, Supplementary Fig. S2). For instance, there was a stronger association between cumulative exposure to a BMI ≥ 25 and/or ≥30 and the risk of colorectal, gallbladder and biliary tract, breast postmenopausal, thyroid, and kidney cancers at lower values of these exposures, after which the increase in risk diminished. For corpus uteri cancer, the risk increased faster than linear at higher values of most exposures. The longitudinal exposures had a similar strength of association with the above mentioned 10 cancer types compared to BMI at index date (in linear models), except for corpus uteri cancer which was stronger for the latter (eg, BMI at index date 1.55 [1.51-1.58] vs cumulative exposure to a BMI ≥ 30: 1.29 [1.27–1.31]) (Fig. 1, Supplementary Table S2). The results of the minimally- and fully adjusted models were similar (Supplementary Fig. S3).

Contrary to BMI at index date, one or more of these longitudinal exposures were also positively associated with the risk of seven cancer types including cancers of the ovary, non-Hodgkin lymphoma, bladder, malignant melanoma of skin, prostate, pancreas, and stomach (Fig. 1, Supplementary Table S2). Duration of BMI ≥ 25 and ≥30, cumulative exposure to a BMI ≥ 25, and age of onset of a BMI ≥ 25 were all positively associated with the risk of non-Hodgkin lymphoma. Duration of BMI ≥ 30 and cumulative exposure to a BMI ≥ 25 and ≥30 were positively related to the risk of ovarian cancer. Longer duration of BMI ≥ 25 and higher cumulative exposure to a BMI ≥ 25 were positively related to the risk of bladder cancer. Although in non-linear analyses only lower levels of cumulative exposure to a BMI ≥ 25 were positively linked to bladder cancer (Fig. 3). Duration of BMI ≥ 25 was further associated with risk of malignant melanoma of the skin, and prostate (for which the association had an attenuated, inverted U-shape in non-linear analyses) cancers (Fig. 3). Age of onset of a BMI ≥ 25 and ≥30 were both related to a higher risk of pancreatic cancer, whereas only BMI ≥ 30 was associated with a greater risk of stomach cancer.

A higher BMI at index date was inversely associated with the risk of six cancer types, of which five were also inversely linked to duration of a BMI ≥ 25 kg/m², including cancers of the stomach and respiratory tract (esophagus [HR, 95% CI: 0.88, 0.82-0.93], larynx, trachea, bronchus, and lung, and head and neck [0.95, 0.92-0.98]) cancers (Fig. 1). These associations were found to be non-linear (Fig. 2 and Supplementary Fig. S2), but while the relationships were L-shaped for BMI at index date, they had an attenuated inverted U-shape for duration of a BMI ≥ 25 (which were similarly shaped for BMI ≥ 30, albeit closer to 1). In addition, although cumulative exposure to a BMI ≥ 25 and/or ≥30 was only inversely related to the risk of cancers of the larynx and trachea, bronchus, and lung in linear analyses, in non-linear models these exposures were related to the risk of stomach and the four respiratory tract cancers in a J-shaped fashion

(Figs. 1 and 3). Age of onset of a BMI ≥ 25 was inversely associated with the risk of larynx cancer.

The results of the supplementary and sensitivity analyses are described in Appendix 1 and reported in Supplementary Figs. S4, S5, S6, S7, S8, S9, S10, and S11. The inverse associations (for stomach and respiratory tract cancers) became null when we restricted the analyses to never smokers. Moreover, BMI at index date, duration of, and cumulative exposure to a BMI ≥ 25 (≥ 30) became positively and more pronouncedly, respectively, associated with head and neck and bladder cancers (Supplementary Fig. S7). Overall, our results were similar to those from four sensitivity analyses although there were some differences in the sensitivity analysis in which we applied the Bonferroni correction (95%CIs changed to 99%) (eg, while no changes were seen for the duration of a BMI ≥ 25 kg/m² exposure between the main and the sensitivity analysis, four out of ten positive associations became null for the age of onset of a BMI ≥ 30 kg/m² exposure) (Supplementary Figs. S10 and S11).

## Discussion

In this population-based cohort study that included 2,645,885 individuals living in Catalonia, Spain, we found that longitudinal BMI-derived exposures and BMI at index date were positively associated with the risk of 12 cancers (corpus uteri, kidney, gallbladder and biliary tract, multiple myeloma, leukemia, breast postmenopausal, colorectal, liver, thyroid, brain and CNS, as well as head and neck and bladder [among never smokers]). Some longitudinal exposures, but not BMI at index date, were additionally positively associated with the risk of six cancer types (ovary, non-Hodgkin lymphoma, malignant melanoma of skin, prostate, pancreas, and stomach cancers). BMI at index date and overweight duration were inversely associated with the risk of stomach and respiratory tract cancers, which likely indicates residual confounding by smoking since these associations were attenuated towards unity when we restricted these analyses to individuals who never smoked. (A table summarizing the main findings of this study can be consulted in Supplementary Fig. S12).

A single measurement of BMI at study baseline has been convincingly associated with the risk of 13 cancer types in previous studies, of which 10 (colorectum, liver, gallbladder and biliary tract, postmenopausal breast, corpus uteri, ovary, kidney, brain and CNS, thyroid, and multiple myeloma) cancers were also positively associated with the longitudinal BMI-derived exposures we investigated (Supplementary Fig. S12)[2]. Thus, our findings seem to indicate that longer exposures to overweight and obesity (with or without accounting for the degree of overweight and obesity), as well as developing overweight and obesity at younger ages in early adulthood might increase cancer risk. This suggests that overweight and obesity prevention should start in early adulthood and that weight management and weight loss interventions leading to shorter durations of overweight and obesity might reduce cancer incidence.

We also provide novel evidence that longitudinal BMI-derived exposures and/or BMI at index date are positively associated with the risk of leukemia, non-Hodgkin lymphoma, malignant melanoma of the skin, prostate, and among never smokers only and more pronouncedly, respectively, with head and neck, and bladder cancers, all of which are not yet considered as obesity-related cancers in the literature (Supplementary Fig. S12)[2]. Furthermore, for some of these cancers (non-Hodgkin lymphoma, malignant melanoma of the skin, prostate, and -in the main analysis- bladder cancers) we only found positive associations for the longitudinal exposures (not for BMI at index date), which highlights that these longitudinal adiposity-related exposures provide additional information compared to a single measure of BMI in time. These additional associations might also indicate that the longitudinal exposures we considered better reflect, than baseline BMI, the underlying biological mechanisms between long-term exposure to adiposity and cancer development.

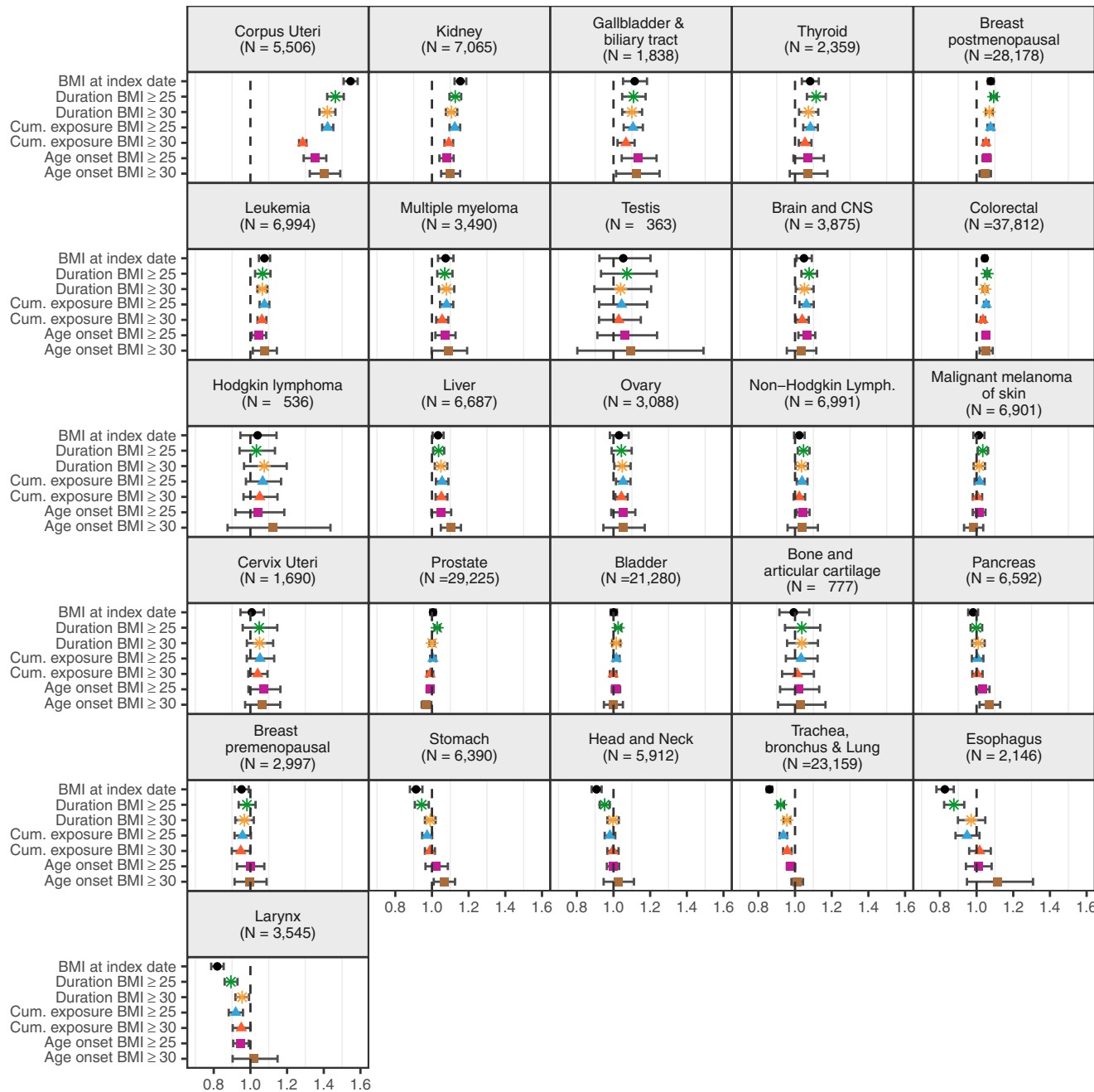

**Fig. 1 | Forest plot of hazard ratios of 26 cancer types related to one standard deviation increment of BMI at index date and other longitudinal BMI-derived exposures, with 95% CIs.** Notes: Data are presented as HRs (per one standard deviation increment) with the respective 95% CIs. Source data are provided as a Source Data file. Models are adjusted for geographic region of nationality, the MEDEA deprivation index, smoking status, and alcohol intake and stratified by age (5-year categories). Cumulative exposure is an exposure considering both degree and duration of overweight/obesity which is obtained by adding the difference between the BMI measurements that were ≥25 (≥30) kg/m² and 24.9 (29.9) kg/m² for every year lived with a BMI ≥25 and ≥30, respectively. Age of onset of a BMI ≥25 (and ≥30) kg/m² is only available for individuals who ever had a BMI ≥25 (≥30) kg/m² (N of cases are in Supplementary Table S5) and the HRs of these exposures were inverted for visualization purposes (ie, an HR >1 means a greater risk at younger ages). Cancer types are ordered by descending ranking of the HRs for BMI at index date. The SD for each exposure were: 10 years for duration of BMI ≥25 and 7 years of BMI ≥30 kg/m², 69 cumulative overweight-years for cumulative exposure to a BMI ≥25 and 36 cumulative obesity-years to a BMI ≥30 kg/m², 7 years for age of onset of a BMI ≥25 and 8 years ≥30 kg/m². Models for ovary, cervix, and corpus uteri cancers were only computed in females, for breast pre-menopausal only in pre-menopausal females, for breast post-menopausal only in post-menopausal females, and for prostate and testis only computed in males (their respective SDs can be consulted in Supplementary Table S3). Brain and CNS includes pituitary gland and pineal gland tumors. Abbreviations: BMI: Body Mass Index; CI: Confidence Interval; CNS: central nervous system; Cum: Cumulative; HR: Hazard Ratio; Lymph: lymphoma.

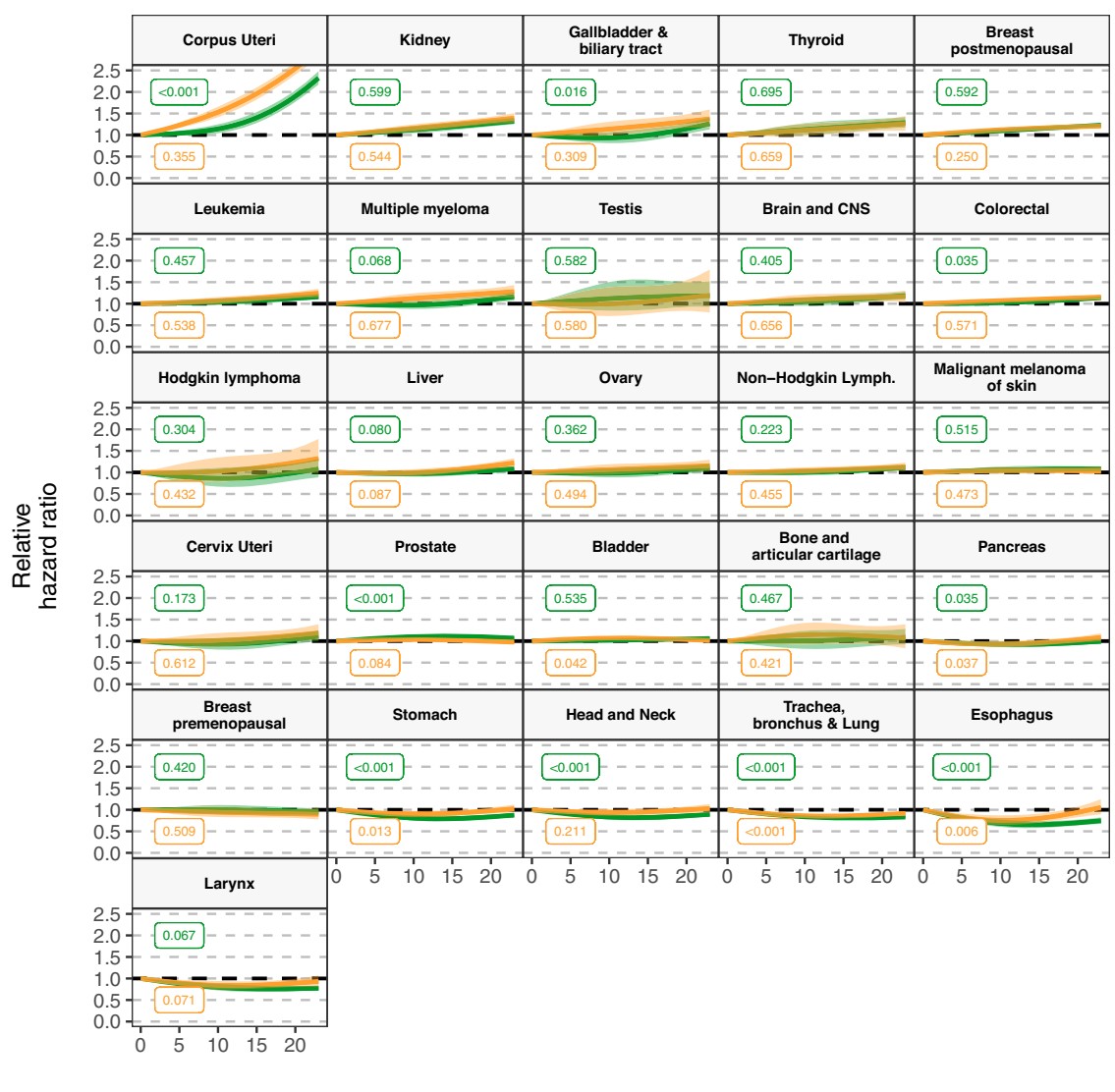

**Fig. 2 | Hazard ratios of 26 cancer types related to duration of BMI ≥ 25 and ≥ 30 kg/m² in years, respectively, with 95% CIs, allowing for non-linearity.** Notes: Data are presented as HRs with the respective 95% CIs. Source data are provided as a Source Data file. Green lines are for the duration of BMI ≥ 25 exposure and yellow for ≥30 kg/m². Models are adjusted for geographic region of nationality, the MEDEA deprivation index, smoking status, and alcohol intake and stratified by age (5-year categories). These graphs were obtained using restricted cubic splines with 3 knots for the exposures of interest with 0 years as the reference point. P-values for nonlinearity were obtained by comparing the model where the exposures were fitted with a nonlinear term against a linear model using a likelihood ratio test (two-sided without adjustment for multiple comparisons). The exact p-values for duration of BMI ≥ 25 were 0.00000004 (for Corpus Uteri), 0.00001 (Prostate), 0.0003 (Stomach), 0.0004 (Head and Neck), 0.000002 (Trachea, bronchus & Lung), and 0.00003 (Esophagus); and for duration of BMI ≥ 30 it was 0.00006 (Trachea, bronchus and Lung). Cancer types are ordered by descending ranking of the HRs for BMI at index date of Fig. 1. Models for ovary, cervix, and corpus uteri cancers were only computed in females, for breast pre-menopausal only in pre-menopausal females, for breast post-menopausal only in post-menopausal females, and for prostate and testis only computed in males. Brain and CNS includes pituitary gland and pineal gland tumors. Abbreviations: BMI Body Mass Index, CI Confidence Interval, CNS central nervous system, Cum Cumulative, Lymph lymphoma.

The IARC viewpoint on excess body fatness and cancer risk considered the evidence as inadequate for leukemia and non-Hodgkin lymphoma (Supplementary Fig. S12)[2]. However, four meta-analyses have reported the association between BMI and higher risk of leukemia and non-Hodgkin lymphoma (or only of diffuse large B cell lymphoma)[3,11–13]. Our findings support and extend these results by providing evidence that higher levels of adiposity through a life course perspective are consistently associated with the risk of hematological cancers, including multiple myeloma, leukemia, and non-Hodgkin lymphoma. Furthermore, we showed that among individuals who never smoked, higher levels of baseline BMI, and longer exposures to overweight and obesity (with or without accounting for the degree of overweight/obesity) are positively associated with the risk of head and

neck and bladder cancers which expands on the extent to which adiposity can affect cancer risk. The three mechanisms by which greater overall adiposity may increase cancer risk have been extensively reported in the literature (sex hormone metabolism, insulin and insulin-like growth factors (IGF) signaling, and adipokine pathways) and could also explain some of the associations between longitudinal BMI-derived exposures and cancer risk (eg, corpus uteri, breast postmenopausal, colorectal cancers)[14–21]. However, other pathways may be involved in the risk of cancer types not yet considered obesity-related and require further research.

Moreover, duration of BMI ≥ 25 kg/m² was the only exposure positively associated with the risk of malignant melanoma of the skin and prostate cancers (Supplementary Fig. S12). This is in line with what

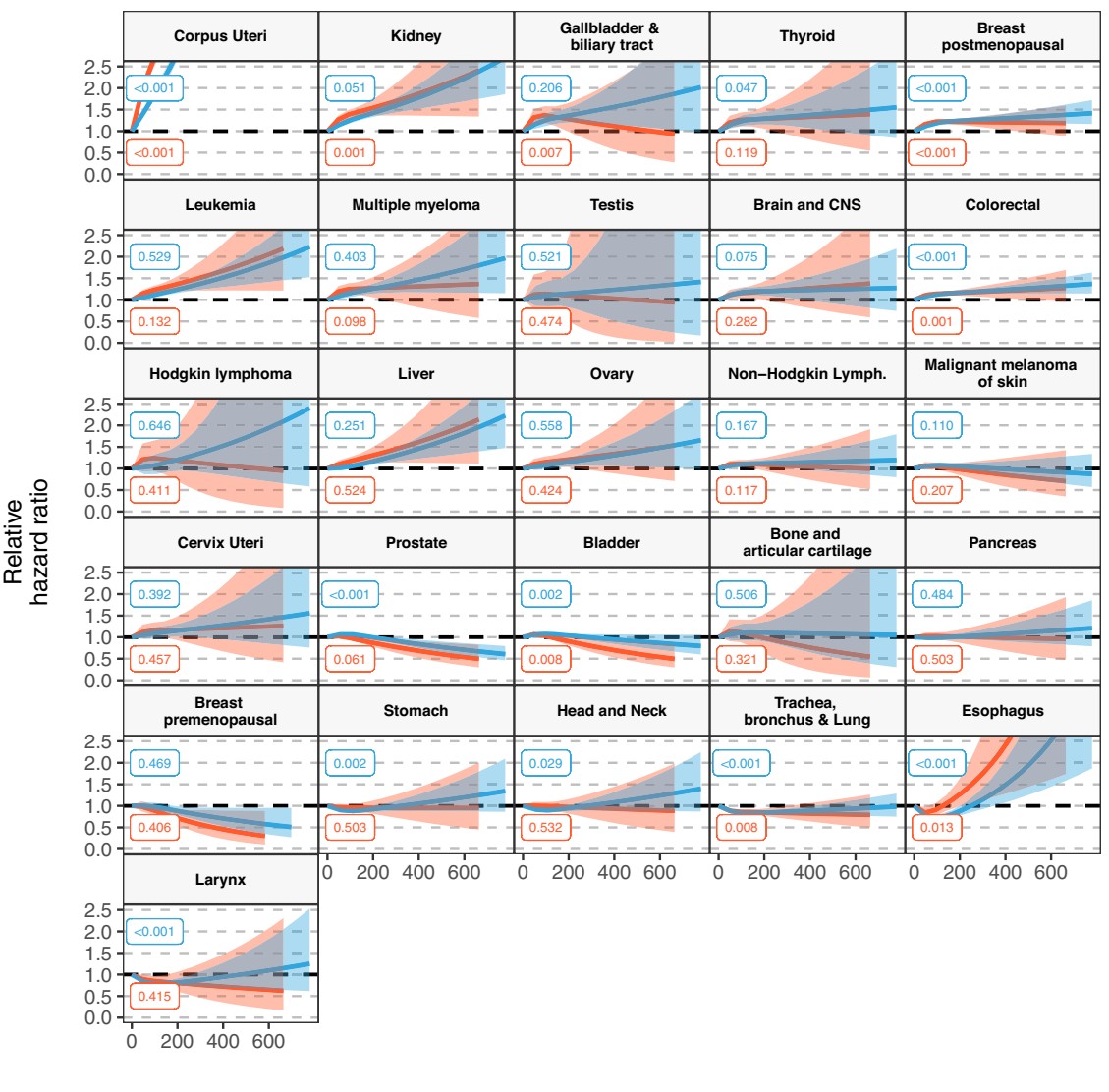

**Fig. 3 | Hazard ratios of 26 cancer types related to cumulative exposure to BMI ≥ 25 and ≥ 30 kg/m² in cumulative overweight and obesity-years, respectively, with 95% CIs, allowing for non-linearity.** Notes: Data are presented as HRs with the respective 95% CIs. Source data are provided as a Source Data file. Blue lines are for the cumulative exposure to BMI ≥ 25 exposure and red for ≥30 kg/m². Models are adjusted for geographic region of nationality, the MEDEA deprivation index, smoking status, and alcohol intake and stratified by age (5-year categories). These graphs were obtained using restricted cubic splines with 3 knots for the exposures of interest with 0 years as the reference point. P-values for nonlinearity were obtained by comparing the model where the exposures were fitted with a nonlinear term against a linear model using a likelihood ratio test (two-sided without adjustment for multiple comparisons). The exact p-values for duration of BMI ≥ 25 were 0.000007 (for Corpus Uteri), 0.0000003 (Breast postmenopausal), 0.0001 (Colorectal), 0.000005 (Prostate), 0.00000002 (Trachea, bronchus and

Lung), 0.00000005 (Esophagus), and 0.0004 (Larynx); and for duration of BMI ≥ 30 they were 0.00000006 (Corpus Uteri) and 0.0000005 (Breast postmenopausal). Cumulative exposure is an exposure considering both degree and duration of overweight/obesity which is obtained by adding the difference between the BMI measurements that were ≥25 (≥30) kg/m² and 24.9 (29.9) kg/m² for every year lived with a BMI ≥ 25 and ≥30, respectively. Cancer types are ordered by descending ranking of the HRs for BMI at index date of Fig. 1. Models for ovary, cervix, and corpus uteri cancers were only computed in females, for breast pre-menopausal only in pre-menopausal females, for breast post-menopausal only in post-menopausal females, and for prostate and testis only computed in males. Brain and CNS includes pituitary gland and pineal gland tumors. Abbreviations: BMI Body Mass Index, CI Confidence Interval, CNS central nervous system, Cum Cumulative, Lymph lymphoma.

was observed in the non-linear analysis of the association between BMI at index date and risk of these cancers (in this and other studies), where an inverted U-shaped association was found, indicating a higher risk of these cancers only for BMIs in the overweight range[22,23]. Future research should focus on confirming these associations and on understanding the pathways by which only being overweight (and a longer duration of it) could have a harmful effect on the risk of these cancers. On the other hand, while higher levels of BMI have been convincingly associated with risk of pancreatic and gastric cardia cancers[2], in our study we only found a positive association with respect

to age of onset of a BMI ≥ 25 ( ≥ 30) kg/m² (Supplementary Fig. S12). The lack of association with stomach cancer for other exposures could be due to our inability to distinguish gastric cardia (obesity-related) from non-cardia cancers (in Spain, the incidence of the non-obesity-related subsite of this cancer is higher than the obesity-related subsite) (Supplementary Fig. S12)[24]. Finally, greater levels of baseline BMI and longer duration of overweight and obesity were inversely associated with risk of respiratory tract cancers in the main analyses, but were attenuated towards unity in the analysis restricted to never smokers (Supplementary Fig. S12). Previous studies have also reported these

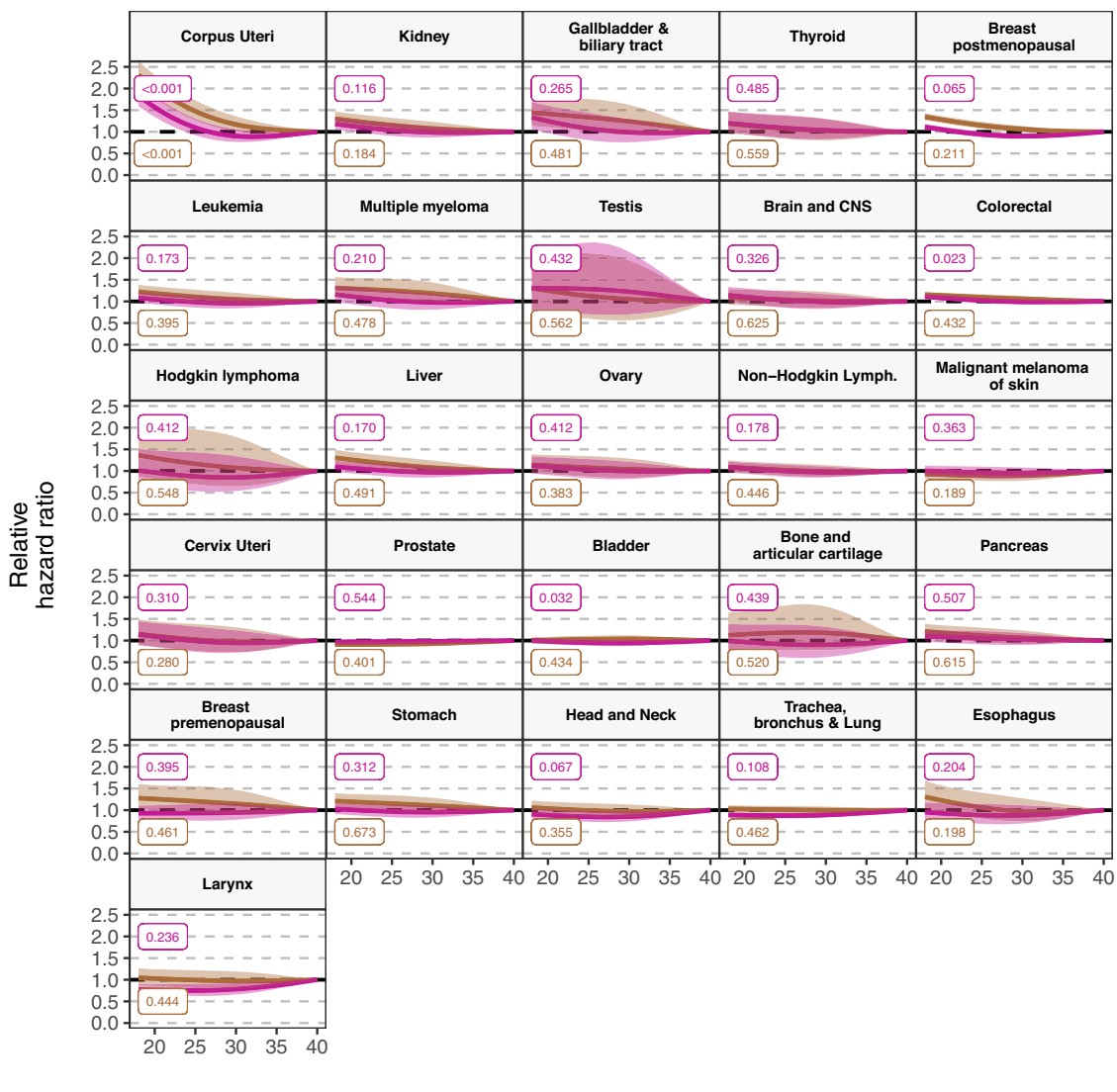

Age of onset BMI ≥ 25            Age of onset BMI ≥ 30

**Fig. 4 | Hazard ratios of 26 cancer types related to age of onset of BMI ≥ 25 and ≥ 30 kg/m² (among people who ever had a BMI ≥ 25 and ≥ 30 kg/m², respectively) in years, respectively, with 95% CIs, allowing for non-linearity.** Notes: Data are presented as HRs with the respective 95% CIs. Source data are provided as a Source Data file. Violet lines are for the age of onset of BMI ≥ 25 exposure and brown for ≥30 kg/m². Models are adjusted for geographic region of nationality, the MEDEA deprivation index, smoking status, and alcohol intake and stratified by age (5-year categories). These graphs were obtained using restricted cubic splines with 3 knots for the exposures of interest with 40 years as the reference point (thus an HR > 1 means a greater risk at younger ages). P-values for nonlinearity were obtained by comparing the model where the exposures were fitted with a nonlinear term against a linear model using a likelihood ratio test (two-sided without adjustment for multiple comparisons). The exact p-values for duration of BMI ≥ 25 were 0.000001 (for Corpus Uteri), and for duration of BMI ≥ 30 it was 0.0001 (Corpus Uteri). Cancer types are ordered by descending ranking of the HRs for BMI at index date of Fig. 1. Models for ovary, cervix, and corpus uteri cancers were only computed in females, for breast pre-menopausal only in pre-menopausal females, for breast post-menopausal only in post-menopausal females, and for prostate and testis only computed in males. Brain and CNS includes pituitary gland and pineal gland tumors. Abbreviations: BMI Body mass index, CI confidence interval, CNS central nervous system; Lymph: Lymphoma.

inverse associations for baseline BMI[22,23], which have hypothesized that this might be due to residual confounding by smoking (lower BMI levels as an approximation of heavy smoking). We hypothesize that the findings for duration of overweight and obesity might also be due to residual confounding by smoking (no or short periods with a BMI ≥ 25 as an approximation of heavy smoking), but further research specifically focussing on this is needed.

This study has several strengths. To our knowledge, this is the first study to analyze the associations between several BMI-derived longitudinal exposures and the risk of numerous (26) cancer types in a single and sufficiently powered data set, including systematic investigation of non-linearity. SIDIAP is representative of the general population of Catalonia in terms of age, sex, and geographic distribution, which lends

external validity to our results[25]. Thanks to the advanced multiple imputation approach for the BMI trajectories, we were able to include all individuals eligible to enter the study, likely minimizing the possibility of selection bias. The diagnoses of the cancer types considered as outcomes have been previously validated and used for BMI and cancer-related research[22,26,27]. While we cannot discard the possibility of outcome misclassification, this was likely not differential according to the exposures, thus, this probably did not greatly affect our results.

Our findings should be interpreted in light of some limitations. The major limitation is that due to the length of follow-up available (12 years), we exclusively relied on multiple imputed BMI measurements for the exposure window. It is important to clarify that the

applied methodology aimed to estimate (not to predict) the BMI of the participants based on the likelihood with other BMI measurements of the participant whose BMI assessments were being imputed as well as other participants' measurements matched on several characteristics. To account for the uncertainty surrounding each BMI estimation, we had 5 imputed BMI trajectories per participant. Nevertheless, the implementation of this approach could have introduced exposure misclassification bias. We aimed to reduce this bias by using high-quality BMI measurements (measured by health professionals and with a distribution shown to be similar to representative studies of the Spanish population)[22,28] and by including data on all adults in SIDIAP ($n = 5,279,567$) for the multiple imputations. We were also empirically reassured about the quality of our exposures given that we found similar associations between BMI-derived longitudinal exposures and risk of specific cancer types that have been previously studied (colorectal, breast postmenopausal, endometrium, kidney, and multiple myeloma cancers)[2,7–10]. Another limitation is that the observed BMI measurements of the participants were very close in time between each other difficulting the capture of granularity (eg, weight cycling) in the trajectories. Also, the dispersion of duration of BMI ≥ 30 and cumulative exposure to a BMI ≥ 30 was modest in the overall population; which could explain why for certain cancers (eg, non-Hodgkin lymphoma or bladder cancers) we only observed statistically-significant associations for the respective exposures of BMI ≥ 25. A disadvantage of the way in which we separated the exposure from the time-to-event window in this study was that it allowed a gap between the two windows. We tried to circumvent this limitation by conducting a sensitivity analysis in which we additionally adjusted our analysis by a variable representing the difference in the BMI at study entry (January 1st, 2009) and that of the age of 40 years (end of the exposure window) which yielded similar results to that of the main analysis. Furthermore, while starting the time-to-event analysis at a specific age (eg, 40 years) instead of a specific date (ie, January 1st, 2009) would have been more congruent with our study design, this would have dramatically reduced the number of participants included in the study and hindered the study of less common malignancies. Another weakness of this study is that we performed multiple tests which could have resulted in false positive associations. Although we were cautious in the interpretation of our main findings (ie, we emphasized consistent associations between the different exposures and a specific cancer type instead of focussing on p-values in isolation), to reduce the likelihood of false positive findings we also implemented the Bonferroni correction (sometimes criticized as being overly conservative) in a sensitivity analysis which revealed consistent findings with our main analysis (Supplementary Fig. S11). Finally, for certain potential confounding factors we had limited information (ie, smoking amount or individual-level SES). While we had access to related indicators such as smoking status or the MEDEA deprivation index, we cannot exclude the possibility of some residual confounding.

In this large Southern European study, we found that longer duration and greater degree of overweight and obesity during early adulthood as well as younger age of onset of a high BMI are associated with a higher risk of 18 cancer types. We provide novel evidence that adiposity over the life course is positively associated with the risk of leukemia, non-Hodgkin lymphoma, as well as head and neck and bladder cancers (among never smokers) and we confirm associations that have been reported in studies focusing on single BMI measurements at study baseline. Our findings reinforce the need for public health strategies for overweight and obesity prevention and reduction in early adulthood for cancer prevention.

## Methods

This study complies with all relevant ethical regulations, including a waiver of individual consent for research use of de-identified electronic health record data. It was approved by the SIDIAP Scientific Committee and the Clinical Research Ethics Committee of the IDIAPJGol (project approval code: P14/074).

### Study design, setting, and data sources

We conducted a population-based cohort study from January 1st, 2009 (index date or baseline date) to December 31st, 2018, using prospectively collected primary care records from the Information System for Research in Primary Care (SIDIAP; www.sidiap.org) in Catalonia, Spain. SIDIAP contains pseudo-anonymized records for >8 million people since 2006[25]. It covers >75% of the population of Catalonia and is representative of the general population of Catalonia by age, sex, and geographic distribution[25]. SIDIAP contains longitudinal data on anthropometric measurements, disease diagnoses (International Classification for Diseases, 10th revision [ICD-10]), sociodemographic and lifestyle information, among others. SIDIAP can be linked to the Minimum Basic Dataset (CMBD), a national population-based registry that includes hospital discharge information of mandatory registration[29].

### Participants

We included 2,645,885 (1,241,523 males and 1,404,362 females) individuals aged ≥40 years (median age was 56 years and the interquartile range was 47 and 68 years) on January 1st, 2009. We excluded individuals without 1 year of history in SIDIAP (to capture their baseline characteristics), and/or with a cancer diagnosis prior to index date (Supplementary Fig. S1). We followed up participants from 1 year after index date (to minimize the possibility of reverse causality [ie, BMI affected by undiagnosed cancer]) until the earliest of cancer diagnosis (any cancer, except other cancer and unspecified malignant neoplasm of the skin), death, transferral out of the SIDIAP catchment area, or end of the study period (December 31st, 2018), whichever occurred first.

### Assessment of variables

To calculate BMI trajectories we extracted data on valid BMI measurements (before applying multiple imputations) assessed from January 1st, 2006 until December 31st, 2018. These were calculated using the weight and height of individuals assessed in a standardized manner by general practitioners or nurses in clinical practice[28]. To be considered valid BMIs, the BMI measurements had to be (i) comprised between 15 kg/m$^2$ and 60 kg/m$^2$ (ie, extremely low values could be indicative of an underlying disease, and extremely low/high values could be due to data entry errors in medical records); (ii) measured at or after 18 years; (iii) not measured during pregnancy (from the 3rd month of pregnancy until 2 months after delivery).

The outcomes were incident diagnoses of 26 cancer types (head and neck; esophagus; stomach; colorectal; liver; gallbladder and biliary tract; pancreas; larynx; trachea, bronchus, and lung; bone and articular cartilage; malignant melanoma of skin; breast [categorized into pre and postmenopausal due to well-established evidence indicating different relations with BMI];[14] cervix uteri; corpus uteri; ovary; prostate; testis; kidney; bladder; brain and central nervous system [CNS]; thyroid; Hodgkin lymphoma; non-Hodgkin lymphoma; multiple myeloma; and leukemia) that have been previously validated in SIDIAP (including the CMBD)[26]. We identified cancer diagnoses using ICD-10 and ICD-9 codes recorded in the SIDIAP and CMBD databases, respectively (Supplementary Table S4).

Potential confounding variables that we were able to consider were age (in 5-year categories) at index date, sex (female, male, as it appears registered in the Catalan public health system), geographic region of nationality (Spanish, Global North, or Global South)[30], the Mortalidad en áreas pequeñas Españolas y Desigualdades Socioeconómicas y Ambientales (MEDEA) deprivation index (an ecological index calculated in urban census tracts, categorized into quintiles by SIDIAP to which we added a rural category since the index was unavailable for participants living in those areas)[31], smoking status (never,

former, or current smoker), and alcohol intake (no, low, or high risk) which is constructed based on type of alcoholic drink, amount, situation, and frequency of consumption[32]. The MEDEA deprivation index was defined based on the census tract where the participants were residing on December 31st, 2018 (date of data extraction). For smoking status and alcohol intake, we selected the closest registry to the index date.

### Statistical analyses

We used a two-step approach for the statistical analyses. Firstly, we estimated life-course BMI trajectories among individuals aged ≥18 years (we excluded those without 1 year of history or follow-up before and after, respectively, their entry into SIDIAP, $n = 5,279,567$). Secondly, we used these trajectories to construct longitudinal BMI-derived exposures among the study participants and we investigated their association with cancer risk using survival models.

**Calculation of BMI trajectories**. We applied multilevel time raster multiple imputation to BMI to obtain the BMI trajectories (five imputed trajectories per individual)[33]. This approach allows to impute irregularly spaced longitudinal data (such as unbalanced BMI assessments in an electronic health records database) relying on within- and between-patient information.

The multilevel component of this approach refers to the imputation model which was a linear mixed model. The cluster variable of this model was each individual's identifier. Level 1 variables (ie, that can vary within individuals) were the age at the BMI measurement and indicator variables of cancer, cardiometabolic conditions (ie, hypertension, type 2 diabetes, and cardiovascular diseases), and bariatric surgery, for which the value was 1 if the individual had been diagnosed with the condition/disease or had the procedure before the BMI assessment (for cancer, 1 year prior), 0 if otherwise (as the diagnosis of these conditions and procedures can lead to changes in BMI). These variables were arbitrarily chosen based on clinical knowledge and due to the quality of the variables' registry in the SIDIAP database. Level 2 variables (ie, that vary between individuals) were sex, the MEDEA deprivation index, smoking status, alcohol intake, geographic region of nationality, the Charlson Comorbidity index (an indicator of multimorbidity composed of 19 items), diagnosis of different cancer types, and follow-up time. These variables were selected based on the multiple imputation literature stating that one should include exposures, covariates, and time of follow up/outcomes (when using a time-to-event analysis) as well as auxiliary variables to impute variables with missing data[34,35].

The time raster component of this approach was used to homogenize the times of BMI measurement. We imputed BMI at six not-equally-spaced age points (18, 30, 40, 55, 70, and 90+ years of age). We used a B-spline of degree 1 to discretize the time of BMI measurement: considering all valid available BMI measurements in an individual's health record, we attributed weights to the spline according to whether the age at measurement of the real BMI measurement coincided with one of the age points of interest. More details on this approach can be consulted in Appendix 2.

To construct the life-course trajectories, we joined two contiguous BMI measurements (ie, between two consecutive age points) with a straight line. This method has previously been used to assess longitudinal changes in BMI in SIDIAP and elsewhere[27,33]. To implement the multilevel time raster multiple imputations we used the library MICE 3.13.0 available for the software R version 4.0.3.

**Calculation of exposures**. We used the BMI trajectories to calculate the exposures and we subsequently analyzed their associations with cancer risk (time-to-event analysis). The window to capture longitudinal exposures was between the ages of 18 and 40 years and was separated from the time-to-event window, which extended from the age of an individual (≥40 years for everyone) one year after index date until the

age at end of follow-up (Supplementary Fig. S13). We generated six longitudinal exposures. The duration of BMI ≥ 25 kg/m² (and of ≥30, respectively) was the sum of years lived with a BMI ≥ 25 (≥30) kg/m². Cumulative exposure to a BMI ≥ 25 kg/m² (and ≥30) was calculated by summing the differences between the BMI measurements that were ≥25 (≥30) kg/m² and 24.9 (29.9) kg/m² for every year lived with a BMI ≥ 25 (≥30) kg/m². For all other years, the value of the cumulative exposure was set to 0[36,37]. Age of onset of a BMI ≥ 25 (and ≥30) kg/m² was the age at which a person had a BMI measurement ≥25 (≥30) kg/m² for the first time in the trajectory and was only available for individuals who ever had a BMI ≥ 25 (≥30) kg/m². Supplementary Fig. S14 shows graphical representations of the exposures. For comparability, we also considered BMI at index date (or at baseline, on January 1st, 2009) as an exposure.

**Association between BMI-derived exposures and cancer risk**. We investigated the association between each of the exposures with the risk of the 26 cancer types by running Cox proportional hazard models with age as the underlying time metric in each of the five imputed datasets and pooling the results using Rubin's rule[38,39]. The minimally-adjusted models included one exposure at a time and were adjusted for sex and stratified by age (5-year categories). The fully adjusted models were further adjusted for the geographic region of nationality, MEDEA deprivation index, smoking status, and alcohol intake. We guided our decisions on the control for confounding by using a directed acyclic graph (Supplementary Fig. S15)[40]. We multiply imputed covariates with missing data at baseline (using predictive mean matching, with 5 imputations drawn) (Appendix 2) and we checked the proportional hazard assumptions for the variables included in the models by visual inspection of survival curves. We calculated the hazard ratios (HRs) and their respective 95% confidence intervals (CIs) for each cancer type per 1 standard deviation (SD) increment of each exposure to allow comparability between the different HRs[41]. We checked whether the 95% CIs of the HR of each longitudinal exposure overlapped with that of BMI at index date to assess differences in the strength of the associations between the longitudinal exposures and BMI at index date. For better interpretability, we inverted the HRs of the models including age of onset as the main exposure (ie, HRs >1 indicate greater risk at younger ages). We also fitted models using restricted cubic splines for the exposures with 3 knots (placed at the 10th, 50th, and 90th percentiles)[42,43]. We evaluated linearity by comparing the difference in log-likelihood of the models with each exposure as a linear and non-linear term.

We conducted five secondary analyses to contextualize our findings. We stratified the analyses by age at index date at two arbitrarily selected age points (<65 or ≥65) and sex. We mutually adjusted the models for the association of age of onset of a BMI ≥ 25 (and ≥30) and duration of BMI ≥ 25 (and ≥30) kg/m² and cancer risk. We restricted the analyses to never smokers to account for possible residual confounding by smoking[44]. We compared the Harrell's C-indices of the models with BMI at index date as the main exposure to the same models further adjusted for each longitudinal exposure separately to assess if the longitudinal exposures improve cancer risk discrimination compared to the standard baseline BMI criterion[43,45]. We recalculated the exposures restricting them to BMIs ≥25 and <30 kg/m² to investigate the independent effect of overweight (from obesity) in relation to cancer risk.

We conducted four sensitivity analyses to assess the robustness of our findings. We (i) further adjusted our models by the difference between the BMI at index date and at 40 years to account for changes in BMI between the start of follow-up and the end of the longitudinal exposure window (see graphical representation in Supplementary Fig. S13), (ii) restricted the analyses to individuals with ≥1 BMI assessment in their health records, (iii) extended the start of the follow-up period from one to 3 years after index date to account for potential reverse causality, and (iv) applied the Bonferroni correction to

counteract the fact that we are testing multiple comparisons. Our confidence intervals were corrected from 95% to 99% [[1-(0.05/6)] x100] considering that we are analyzing six novel exposures (BMI at index date was included for comparability purposes) and cancer risk (each cancer type being considered as a different and specific disease).

## Reporting summary

Further information on research design is available in the Nature Portfolio Reporting Summary linked to this article.

## Data availability

The main data supporting the findings of this study are available within the article and its Supplementary information. The source data underlying figures and Supplementary Figures are provided as Source Data File. Additional details on datasets (such as aggregated data) and protocols that support findings of this study will be made available by the corresponding authors upon request. The authors will give feedback within 30 days. The raw data used in this study cannot be deposited in an online database and are only available for the researchers participating in this study in accordance with the data extraction agreement with SIDIAP and with current European and national law. The General Data Protection Regulation (GDPR) is a key piece of legislation in Europe that governs the collection, processing, and storage of personal data, including electronic health records (EHRs) (such as SIDIAP, the database underlying this study). The GDPR imposes strict requirements on organizations that handle personal data, including requirements for obtaining consent, providing transparency, and implementing appropriate technical and organizational measures to protect data. Given the sensitive nature of EHRs and the stringent requirements for their handling under GDPR, making a database of EHRs such as SIDIAP publicly available would be a violation of privacy laws and could result in serious consequences for the organization responsible for the breach. Additionally, these could represent risks to individual privacy and security, as the data could be accessed, misused, or stolen by unauthorized third parties. However, researchers from public institutions can request data from SIDIAP. The specific conditions for data access are available online (https://www.sidiap.org/index.php/en/solicituds-en) or by contacting the SIDIAP team (sidiap@idiapjgol.org). Source data are provided with this paper.

## Code availability

The analytical code used in this study is available at https://github.com/andrepist/LongitudinalBMIandCancerRisk.

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

## Acknowledgements

We would like to thank all healthcare professionals of Catalonia who daily register information in the populations' electronic health records. Funding was obtained from Wereld Kanker Onderzoek Fonds (WKOF grant number: 2017/1630, awarded to HF), as part of the international grants from the World Cancer Research Fund. MR's salary was also funded by World Cancer Research Fund (UK) (grant number: IIG_2019_1978, awarded to TDS), as part of the World Cancer Research Fund International grant program. TDS acknowledges receiving financial support from the Instituto de Salud Carlos III (ISCIII; Miguel Servet 2021: CP21/00023). The funders had no role in study design, data collection, analysis, decision to publish, or preparation of the manuscript. Where authors are identified as personnel of the International Agency for Research on Cancer and World Health Organization, the authors alone are responsible for the views expressed in this Article and they do not necessarily represent the decisions, policy, or views of the International Agency for Research on Cancer and World Health Organization.

## Author contributions

M.R. performed the literature review. A.P. did the data management and data analysis with contributions from all authors. M.R. wrote the first draft with insightful contributions from A.P., H.F., and T.D.-S. All authors were involved in the study conception and design, interpretation of the results, manuscript preparation, and approved the final version of the manuscript.

## Competing interests

The authors declare no competing interests.

## Ethical approval

This study complies with all relevant ethical regulations, including a waiver of individual consent for research use of de-identified electronic health record data. It was approved by the SIDIAP Scientific Committee and the Clinical Research Ethics Committee of the IDIAPJGol (project approval code: P14/074).
