## [Peer Review File · Nature Communications]

Reviewer comments, first round -

Reviewer #1 (Remarks to the Author):

Recalde and Pistillo et. al. used longitudinal data from a population-based cohort (2009-2018) of electronic health records in Catalonia, Spain, to investigate the association of baseline BMI (2009) and 6 longitudinal BMI-derived exposure calculated between age 18 and 40 years with incident cancer at 26 anatomical sites. More than 2.6 million participants who were free of cancer and aged ≥ 40 years were enrolled in the study in 2009, with 9% of them diagnosed with cancer during the follow-up. The authors found that baseline BMI, duration of, cumulative exposure to, and age of onset of overweight ($\geq 25\text{kg/m}^2$) and obesity ($\geq 30\text{kg/m}^2$) were positively associated with the cancers of corpus uteri, kidney, gallbladder, thyroid, breast (postmenopausal), brain, leukemia, multiple myeloma, colorectal, liver. Longitudinal exposures were also positively associated with non-Hodgkin lymphoma, malignant melanoma of the skin, and ovary, prostate, pancreas, and stomach cancers. The robustness of these results was examined using multiple sensitivity analyses. The authors concluded that their findings support public health strategies of preventing and reducing early overweight and obesity to prevent cancer.

Overall, this manuscript is well-written and may provide additional evidence for the associations of the duration, degree, and age of onset of overweight and obesity during early adulthood with cancer risk.

Major comments:

1. Abstract: The participants were already aged ≥ 40 years at the study baseline in 2009. How were their BMI during 18-40 years obtained? If the true baseline was at age 18 years, calling 2009 the baseline might be confusing.
2. Did the authors have cancer information from age 18 years? Because BMI information since age 18 years were available, why only analyzed incident cancer after 2009? To avoid the scenario depicted by participant 2 in Figure S2 (a gap between longitudinal BMI exposure at 18-40 years and baseline BMI at 60 years), have the authors considered following participants from a certain age instead of an exact date (i.e., 01.01.2009)?
3. An explanation of why a multilevel time raster multiple imputation of BMI was used may be needed. Did the cohort have a lot of missing BMI data? The amount of missing data should be provided.
4. A total of 7 BMI exposures and 26 cancer outcomes were tested. Have the authors considered controlling for multiple hypothesis testing using methods like Bonferroni or Benjamini-Hochberg?

Minor comments:

1. Line 84: Citation is missing after the sentence "Few studies have investigated ...".
2. BMI trajectory usually depicts the different trends of BMI across years (e.g., normal stable, normal to overweight, chronic obesity). "BMI history" may be a more appropriate term to describe the longitudinal BMI measures in this study.

Reviewer #2 (Remarks to the Author):

This manuscript describes associations between BMI trajectories and cancer risk in a large population and adds new information to the literature about the contribution of the duration of overweight and obesity to incident cancer. The methods section regarding the source of the exposure data is not clear. Thus, either, the data prior to 2006 were completely imputed which is a design flaw, or the methods section is not clear and simply needs more clarification.

It seems that BMI was imputed for timepoints prior to 2006 as the database started in 2006. For example, on page 4, line 119, the authors indicate that they included persons aged 40 years of age or older on January 1, 2009. Yet, on page 5, line 150, they indicated they estimated BMI trajectories among individuals 18 year of ages or older. If the SIDIAP started in 2006, where were the BMI values obtained from for the years preceding 2006? Were they all imputed? This is confusing and needs to be clarified.

On page 5, lines 139-147. It is not clear if the potential confounders considered were time varying or captured only at baseline or last follow-up.

Reviewer #3 (Remarks to the Author):

The manuscript under consideration evaluates the relationship between a number of measures of adiposity (including baseline BMI and six longitudinal BMI-derived exposures for overweight and obesity [duration of, cumulative exposure to, and age of onset of BMI \geq 25 and \geq 30kg/m², respectively]) and the incidence of 26 different cancer types in a population-based cohort of 2.6 million adults in Spain. BMI measures were abstracted from medical records. Multiple imputation was used to calculate weight trajectories between ages 18 and 40. The authors identified a relationship between both baseline BMI and longitudinal measures of adiposity in 10 malignancies (uterine, kidney, gallbladder, thyroid, breast (postmenopausal), brain, leukemia, multiple myeloma, colorectal, and liver and, among never smokers, with head and neck and bladder cancers. Longitudinal exposures, but not baseline BMI, were positively associated with the risk of a number of additional malignancies (non-Hodgkin lymphoma, melanoma, and ovary, prostate, pancreas, and stomach cancers).

Although many prior publications have focused on the relationship between measures of adiposity (including longitudinal measures of adiposity) and cancer risk, the novelty of this manuscript lies in the fact that it reports on relationships between cancer risk and objectively collected measures of BMI and that the data set is large enough to provide sufficient power to evaluate the relationship between adiposity and less common malignancies. The authors also conducted a number of sensitivity analyses to further explore their findings. The manuscript is well written, although the results are a bit dense and might benefit from a table in addition to the figures presented to better depict the many relationships reported.

My major concern is that the authors have created two reference categories of adiposity: BMI \geq 25kg/m² and BMI \geq 30kg/m². Many other publications have demonstrated a relationship between BMI \geq 30kg/m², but there are fewer data suggesting that a BMI in the 25-30kg/m² range is associated with an increased risk of cancer. Given the size of the data set, it would be helpful to better characterize whether patients with a BMI in the overweight range are at increased risk of malignancy, vs this excess risk being driven by increased risk in individuals with a BMI in the obese range.

How was pregnancy accounted for in the weight trajectories? Given that more than 50% of the individuals in this cohort were women of childbearing years, pregnancy weights could confound these analyses given the use of multiple imputation to calculate missing weights (especially since women may be more likely to seek medical care and thus have recorded weights during pregnancy).

It would also be helpful to understand how many weights were available for individuals included in these analyses. The supplemental materials include a detailed description of the imputation methods used to calculate missing weights, but there is no information about the average number of weight measurements available. It is helpful that sensitivity analyses omitting individuals with just one weight did not show significantly different outcomes, but it would be helpful to understand how much of the data reported were actual measurements vs imputed values.

Were any adjustments made for multiple testing? The authors report outcomes for 26 cancers using many different measures of weight at particular timepoints and over time. The risk of false positive results is particularly concerning given the narrowly significant associations in some malignancies and the finding in others that some adiposity measures were significantly associated with cancer risk while others were not (for example in melanoma and ovarian cancer). At the very least, this needs to be addressed in the limitations section of the manuscript.

Manuscript ID: NCOMMS-22-14260-T

Title of the article: Longitudinal body mass index-derived exposures during early adulthood and risk of 26 types of cancer: a cohort study of 2.6 million adults in Catalonia, Spain

We are grateful to the Reviewers for the time and expertise invested in these reviews. We have answered all comments and implemented the suggestions made in the manuscript when appropriate. A list of the numbered detailed responses is given below with the following format:

- **Black (bold): a copy of the text written by the Reviewer**
- Blue (normal): our response
- *Blue (italic): a copy of the modified text from the manuscript*

We hope our responses fully addressed the Reviewers' concerns and we wish to submit a revised version of the manuscript for further consideration in this journal.

Looking forward to the outcome of your assessment.

Yours sincerely,

Dr. Talita Duarte-Salles, Fundació Institut Universitari per a la recerca a l'Atenció Primària de Salut Jordi Gol i Gurina (IDIAPJGol), Barcelona, Spain

Dr. Heinz Freisling, International Agency for Research on Cancer-World Health Organization, Lyon, France

Dr. Veronica Davila-Batista, Endocrinology Department, Complejo Hospitalario Universitario Insular Materno Infantil, Las Palmas de GC, Canary Health Service, Las Palmas, Spain

Reviewer #1 (Remarks to the Author):

Recalde and Pistillo et. al. used longitudinal data from a population-based cohort (2009-2018) of electronic health records in Catalonia, Spain, to investigate the association of baseline BMI (2009) and 6 longitudinal BMI-derived exposure calculated between age 18 and 40 years with incident cancer at 26 anatomical sites. More than 2.6 million participants who were free of cancer and aged ≥ 40 years were enrolled in the study in 2009, with 9% of them diagnosed with cancer during the follow-up. The authors found that baseline BMI, duration of, cumulative exposure to, and age of onset of overweight ($\geq 25\text{kg/m}^2$) and obesity ($\geq 30\text{kg/m}^2$) were positively associated with the cancers of corpus uteri, kidney, gallbladder, thyroid, breast (postmenopausal), brain, leukemia, multiple myeloma, colorectal, liver. Longitudinal exposures were also positively associated with non-Hodgkin lymphoma, malignant melanoma of the skin, and ovary, prostate, pancreas, and stomach cancers. The robustness of these results was examined using multiple sensitivity analyses. The authors concluded that their findings support public health strategies of preventing and reducing early overweight and obesity to prevent cancer.

Overall, this manuscript is well-written and may provide additional evidence for the associations of the duration, degree, and age of onset of overweight and obesity during early adulthood with cancer risk.

We thank the Reviewer for the time spent and insights provided to our manuscript. We hope we were able to clarify the corresponding aspects of the manuscript mentioned in the comments below.

Major comments:

1. Abstract: The participants were already aged ≥ 40 years at the study baseline in 2009. How were their BMI during 18-40 years obtained? If the true baseline was at age 18 years, calling 2009 the baseline might be confusing.

We used a two-step approach for the statistical analyses of the present study. Firstly, we estimated life-course BMI trajectories among individuals aged ≥ 18 years (we excluded those without one year of history or follow-up before and after, respectively, their entry into SIDIAP, $n=5,279,567$). We have provided a detailed explanation of how we conducted this 1st step (we used the multilevel time raster multiple imputation approach to obtain the BMI trajectories) to address the 3rd comment of the Reviewer. We hope that the modification of the manuscript as described in Comment 3 (and Comment 1 of the 2nd Reviewer) helps to understand better how we obtained BMIs from 18 to 40 years of age. Then, as a second step, we used these trajectories to construct longitudinal BMI-derived exposures among the study participants ($n=2,645,885$) and we investigated their association with cancer risk using survival models. In this step, we only focused on the part of the BMI trajectories between the ages of 18 and 40 years (the window to capture longitudinal exposures) (graphically represented in Figure S3 of the supplementary material).

When we refer to BMI at baseline we refer to that of January 1st, 2009, therefore the one the participant had at the start of the follow-up or time-to-event window (at age ≥ 40 years). Considering that this is a study based on electronic health records in which data is registered continuously, a baseline date definition is required. In this study, we had six longitudinal exposures and “BMI at baseline” (or “BMI at index date”, as of January 1st, 2009). The latter was included for comparability with studies investigating the association between adiposity and cancer risk which tend to focus on single BMI measurements assessed at study baseline or start of follow-up, which are measures of current (or proxy)

BMI status. That is why “baseline BMI” is not the one measured at 18 years but at the age that the person had at the study entry. We have added a note between parentheses at the beginning of the abstract to help to clarify this:

- *Whether life course adiposity-related exposures are more relevant cancer risk factors than baseline BMI (ie, at start of follow-up for disease outcome) remains unclear.*

2. Did the authors have cancer information from age 18 years? Because BMI information since age 18 years were available, why only analyzed incident cancer after 2009? To avoid the scenario depicted by participant 2 in Figure S2 (a gap between longitudinal BMI exposure at 18-40 years and baseline BMI at 60 years), have the authors considered following participants from a certain age instead of an exact date (i.e., 01.01.2009)?

We understand the Reviewers’ concerns regarding the definition and separation of the exposure and time-to-event windows. In our study, the exposures were calculated using the BMI trajectories during early adulthood (18 until 40 years) which allowed every study participant to obtain the same duration time of exposure (thus permitting the same interpretation of the longitudinal exposures across participants) that was subsequently linked to cancer risk from January 1st, 2009 onwards (≥ 40 years). Having the same exposure window is especially important for cumulative exposures as every participant has the same probability of accumulating duration or degree of overweight/obesity. By design, every study participant could also be potentially followed-up for the same amount of time (from January 1st, 2009 until December 31st, 2018). The advantage of this approach was to separate the two windows of exposure and risk (of having cancer), likely reducing the possibility of bias related to the potential overlap between the 2 (e.g., someone with an event would by definition have a shorter exposure period). Another strength is that this type of approach has been implemented in other studies of the field (1-4) and is easier to interpret than other types of techniques as it mimics a traditional cohort study (with an exposure measured at a point in time prior to -and explicitly separated from- the time-to-event window). We did consider other types of approaches such as joint modeling for longitudinal data to avoid this window segmentation, but this methodology was problematic when trying to implement longitudinal multiple imputations. A disadvantage of the approach that we used in this study is that it allows a gap between the two windows, as correctly pointed out by the Reviewer. We tried to circumvent this limitation by conducting a sensitivity analysis in which we additionally adjusted our analysis by a variable representing the difference in the BMI at study entry (January 1st, 2009) and that of the age of 40 (end of the exposure window) which yielded similar results to that of the main analysis.

We agree with the Reviewer that picking a specific age instead of a date to indicate the beginning of the time-to-event window would be more straightforward. Although we had information on cancer diagnoses from the age of 18 years onwards, our main constraint for choosing an arbitrary age to start the follow-up was that SIDIAP systematically collects incident cancer cases (and all the data available in the database) since January 1st, 2006. Therefore, selecting an arbitrary exact age between 2006 and 2018 would have dramatically reduced the number of participants included in the study and would have resulted in a less heterogeneous population (e.g., if the selected age would have been 40 years, then we would have had a younger population). Apart from that, we could have also considered other arbitrary dates instead of January 1st, 2009, but we thought that this date was an acceptable trade-off between having individual data (BMI measurements and variables used for the multilevel time raster multiple imputations as well as covariates for the Cox models) measured prior to the start of the time-to-event period and sufficient follow-up time to observe incident cancer diagnoses.

We have modified the limitations section of the manuscript to address these issues (page 8, paragraph 2; page 9, paragraph 1):

- *A disadvantage of the way in which we separated the exposure from the time-to-event window in this study was that it allowed a gap between the two windows. We tried to circumvent this limitation by conducting a sensitivity analysis in which we additionally adjusted our analysis by a variable representing the difference in the BMI at study entry (January 1st, 2009) and that of the age of 40 years (end of the exposure window) which yielded similar results to that of the main analysis. Furthermore, while starting the time-to-event analysis at a specific age (eg, 40 years) instead of a specific date (ie, January 1st, 2009) would have been more congruent with our study design, this would have dramatically reduced the number of participants included in the study and hindered the study of less common malignancies.*

References

1. Arnold M, Charvat H, Freisling H, et al. Adult Overweight and Survival from Breast and Colorectal Cancer in Swedish Women. *Cancer Epidemiol Biomarkers Prev.* 2019 Sep;28(9):1518-1524. doi: 10.1158/1055-9965.EPI-19-0075.
2. Arnold M, Freisling H, Stolzenberg-Solomon R, et al. Overweight duration in older adults and cancer risk: a study of cohorts in Europe and the United States. *European Journal of Epidemiology.* 2016;31(9):893-904. doi:10.1007/s10654-016-0169-z
3. Noh H, Charvat H, Freisling H, et al. Cumulative exposure to premenopausal obesity and risk of postmenopausal cancer: A population-based study in Icelandic women. *International Journal of Cancer.* 2020;147(3):793-802. doi:https://doi.org/10.1002/ijc.32805
4. Arnold M, Jiang L, Stefanick ML, et al. Duration of Adulthood Overweight, Obesity, and Cancer Risk in the Women's Health Initiative: A Longitudinal Study from the United States. *PLOS Medicine.* 2016;13(8):e1002081-. <https://doi.org/10.1371/journal.pmed.1002081>

3. An explanation of why a multilevel time raster multiple imputation of BMI was used may be needed. Did the cohort have a lot of missing BMI data? The amount of missing data should be provided.

To address the Reviewer's valid comment, we have provided a more detailed explanation of why we used the multilevel time raster multiple imputation approach for the BMI trajectories (page 11; paragraph 4; page 12; paragraphs 1-3):

- *Calculation of BMI trajectories*
We applied multilevel time raster multiple imputation to BMI to obtain the BMI trajectories.¹⁹ This approach allows to impute irregularly spaced longitudinal data (such as unbalanced BMI assessments in an electronic health records database) relying on within- and between-patient information.
The "multilevel" component of this approach refers to the imputation model which was a linear mixed model. The cluster variable of this model was each individual's identifier. Level 1 variables (ie, that can vary within individuals) were the age at the BMI measurement and indicator variables of cancer, cardiometabolic conditions (ie, hypertension, type 2 diabetes, and cardiovascular diseases), and bariatric surgery, for which the value was 1 if the individual had been diagnosed with the condition/disease or had the procedure before the BMI assessment (for cancer, 1 year prior), 0 if otherwise (as the diagnosis of these conditions and these

procedures can lead to changes in BMI). Level 2 variables (ie, that vary between individuals) were sex, the MEDEA deprivation index, smoking status, alcohol intake, geographic region of nationality, the Charlson Comorbidity index (an indicator of multimorbidity composed of 19 items), diagnosis of different cancer types, and follow-up time.

The “time raster” component of this approach was used to homogenize the times of BMI measurement. We imputed BMI at six not-equally-spaced age points (18, 30, 40, 55, 70, and 90+ years of age). We used a B-spline of degree 1 to discretize the time of BMI measurement: considering all valid available BMI measurements in an individual’s health record, we attributed weights to the spline according to whether the age at measurement of the real BMI measurement coincided with one of the age points of interest. More details on this approach can be consulted in Appendix 1.

To construct the life-course trajectories, we joined two contiguous BMI measurements (ie, between two consecutive age points) with a straight line. This method has previously been used to assess longitudinal changes in BMI in SIDIAP and elsewhere.^{19,20} To implement the multilevel time raster multiple imputations we used the library MICE 3.13.0 available for the software R version 4.0.3.

To address the comment on missing data, we added a supplementary table (Table S2) describing the baseline characteristics of the study population with and without a BMI assessment and we described this table in the results (page 3, paragraph 3) section of the manuscript:

- *Of the included participants, 2,081,840 (79%) had at least one BMI assessment in their electronic health records while 564,045 (21%) had none (Table S2). Individuals without a BMI assessment were more likely to be men and younger compared to those with a BMI assessment. The former also had a higher proportion of non-Spanish, individuals living in the least deprived areas of Catalonia and people with no comorbidities, compared to the latter (Table S2).*

4. A total of 7 BMI exposures and 26 cancer outcomes were tested. Have the authors considered controlling for multiple hypothesis testing using methods like Bonferroni or Benjamini-Hochberg?

This is a very important point. In the first version of the manuscript that we submitted, we decided not to adjust for multiple testing because we had a pre-specified hypothesis (adiposity may be related to more cancer types than currently recognized in the adiposity-cancer literature). From our point of view, our study differs from others that are solely interested in statistically-significant associations between different unrelated and numerous exposures/outcomes in the sense that our different exposures capture the same concept (adiposity) and that the different incident cancer types can be considered as distinct diseases. This being said, we understand the Reviewer’s concern on this topic and for such reason, we have added a sensitivity analysis in which we applied the Bonferroni correction to counteract the multiple comparisons problem. Our confidence intervals were corrected from 95% to 99% [$1-(0.05/6)$] considering that the hypotheses that we are mainly interested in involve the association between the 6 longitudinal exposures and the risk of different cancer types (not on BMI at baseline -the 7th exposure- that has been widely analyzed in the literature). This sensitivity analysis is mentioned in the methods and results sections (page 14, paragraph 2 and page 5, paragraph 2) of the manuscript as well as in appendix 2 (where we described the results of the secondary and sensitivity analyses):

- *We conducted four sensitivity analyses to assess the robustness of our findings. We [...] iv) applied the Bonferroni correction to counteract the fact that we are testing multiple*

comparisons. Our confidence intervals were corrected from 95% to 99% $[[1-(0.05/6)] \times 100]$ considering that we are analyzing six novel exposures (BMI at index date was included for comparability purposes) and cancer risk (each cancer type being considered as a different and specific disease).

- *Overall, our results were similar to those from four sensitivity analyses although there were some differences in the sensitivity analysis in which we applied the Bonferroni correction (95% CIs changed to 99%) (eg, while no changes were seen for the duration of a BMI ≥ 25 kg/m² exposure between the main and the sensitivity analysis, four out of ten positive associations became null for the age of onset of a BMI ≥ 30 kg/m² exposure) (Figure S13 and S14).*
- *Further, when we applied the Bonferroni correction to counteract the fact that we were testing multiple comparisons, some of the positive associations of the main analysis became null (Figure S14). Longer duration of a BMI ≥ 30 kg/m² was positively associated with the risk of 9 cancers in the sensitivity analysis instead of 12 in the main analysis (affecting non-Hodgkin lymphoma and cancers of the brain and the CNS and liver). Higher cumulative exposure to a BMI ≥ 25 kg/m² with 12 instead of 13 cancers (no longer with bladder cancer). Higher cumulative exposure to a BMI ≥ 30 kg/m² with 10 instead of 11 cancers (no longer with brain and the CNS cancer). Age of onset of a BMI ≥ 25 kg/m² with 8 instead of 11 cancers (no longer with leukemia, and cancers of the liver and pancreas). Age of onset of a BMI ≥ 30 kg/m² with 6 instead of 10 cancers (no longer with leukemia, multiple myeloma, and cancers of the stomach and gallbladder and biliary tract). Finally, BMI at index date with 8 instead of 10 cancers (no longer with cancers of the brain and the CNS and liver) (Figure 1, Figure S14).*

The findings of this analysis are presented in Figure S14 of the supplementary material.

Furthermore, we expanded the limitations section of the manuscript to address the multiple testing issue (page 9, paragraph 1):

- *Another weakness of this study is that we performed multiple tests which could have resulted in false positive associations. Although we were cautious in the interpretation of our main findings (ie, we emphasized consistent associations between the different exposures and a specific cancer type instead of focussing on p-values in isolation), to reduce the likelihood of false positive findings we also implemented the Bonferroni correction (sometimes criticized as being overly conservative) in a sensitivity analysis which revealed overall consistent findings (with some exceptions) with our main analysis (Figure S14).*

Finally, we have tried to be cautious about the interpretation of the findings of this study and emphasized consistent associations between the different exposures and a specific cancer type instead of focussing on p-values in isolation. For instance, although we found positive associations between the different exposures and 18 cancer types, four cancer types were only linked to one exposure (malignant melanoma of the skin and cancers of the prostate, pancreas, and stomach) which was cautiously reasoned in the discussion section (page 7, paragraph 2):

- *Moreover, duration of BMI ≥ 25 kg/m² was the only exposure positively associated with the risk of malignant melanoma of the skin and prostate cancers (Figure S15). This is in line with what was observed in the non-linear analysis of the association between BMI at index date and risk of these cancers (in this and other studies), where an inverted U-shaped association was found, indicating a higher risk of these cancers only for BMIs in the overweight range.^{40,41} Future research should focus on confirming these associations and on understanding the pathways by which only being overweight (and a longer duration of it) could have a harmful effect on the risk of these cancers. On the other hand, while higher levels of BMI have been convincingly*

associated with risk of pancreatic and gastric cardia cancers,² in our study we only found a positive association with respect to age of onset of a BMI \geq 25 (\geq 30) kg/m² (Figure S15). The lack of association with stomach cancer for other exposures could be due to our inability to distinguish gastric cardia (obesity-related) from non-cardia cancers (in Spain, the incidence of the non-obesity-related subsite of this cancer is higher than the obesity-related subsite) (Figure S15).⁴²

Minor comments:

1. Line 84: Citation is missing after the sentence “Few studies have investigated ...”.

We thank the Reviewer for spotting this. We added the references “6-10” to that sentence (page 2, paragraph 2):

- *Few studies have investigated the association between longitudinal BMI-derived exposures and cancer risk.⁶⁻¹⁰*

2. BMI trajectory usually depicts the different trends of BMI across years (e.g., normal stable, normal to overweight, chronic obesity). “BMI history” may be a more appropriate term to describe the longitudinal BMI measures in this study.

We appreciate the Reviewer’s comment. We agree that some authors have referred to latent-class group-based trajectory models to define groups of individuals with similar patterns of BMI change during adulthood (eg, stable normal, normal to overweight, etc) (1-3). However, the term “BMI trajectories” has also been used to describe methodological approaches (involving mixed-effects models) similar to ours (4-6). More specifically, the methodological background of the multilevel time raster multiple imputation approach (which was used in this study), also refers to the longitudinal analysis of data (eg, BMI) by the term “trajectories” (7). Therefore, we believe that despite the different use of the term “BMI trajectories” in the literature, the use that we are giving to it in this study is valid.

References

1. Kelly SP, Graubard BI, Andreotti G, Younes N, Cleary SD, Cook MB. Prediagnostic Body Mass Index Trajectories in Relation to Prostate Cancer Incidence and Mortality in the PLCO Cancer Screening Trial, JNCI: Journal of the National Cancer Institute. 2017;109(3):djw225, <https://doi.org/10.1093/jnci/djw225>
2. Nagin DS, Odgers CL. Group-based trajectory modeling in clinical research. *Annu Rev Clin Psychol.* 2010;6:109–138.
3. Kelly SP, Lennon H, Sperrin M, Matthews C, Freedman ND, Albanes D, et al. Body mass index trajectories across adulthood and smoking in relation to prostate cancer risks: The NIH-AARP diet and health study. *International Journal of Epidemiology.* 2019;48(2), 464–473.
4. Arnold M, Freisling H, Stolzenberg-Solomon R, et al. Overweight duration in older adults and cancer risk: a study of cohorts in Europe and the United States. *European Journal of Epidemiology.* 2016;31(9):893-904. doi:10.1007/s10654-016-0169-z
5. Noh H, Charvat H, Freisling H, et al. Cumulative exposure to premenopausal obesity and risk of postmenopausal cancer: A population-based study in Icelandic women. *International Journal of Cancer.* 2020;147(3):793-802. doi:<https://doi.org/10.1002/ijc.32805>

6. Arnold M, Jiang L, Stefanick ML, et al. Duration of Adulthood Overweight, Obesity, and Cancer Risk in the Women's Health Initiative: A Longitudinal Study from the United States. *PLOS Medicine*. 2016;13(8):e1002081-. <https://doi.org/10.1371/journal.pmed.1002081>
7. van Buuren S. *Chapter 11.3: Time raster imputation from Flexible Imputation of Missing Data: Time Raster Imputation*. 2nd Edition. Chapman & Hall/CRC; 2012.

Reviewer #2 (Remarks to the Author):

This manuscript describes associations between BMI trajectories and cancer risk in a large population and adds new information to the literature about the contribution of the duration of overweight and obesity to incident cancer. The methods section regarding the source of the exposure data is not clear. Thus, either, the data prior to 2006 were completely imputed which is a design flaw, or the methods section is not clear and simply needs more clarification.

It seems that BMI was imputed for timepoints prior to 2006 as the database started in 2006. For example, on page 4, line 119, the authors indicate that they included persons aged 40 years of age or older on January 1, 2009. Yet, on page 5, line 150, they indicated they estimated BMI trajectories among individuals 18 year of ages or older. If the SIDIAP started in 2006, where were the BMI values obtained from for the years preceding 2006? We they all imputed? This is confusing and needs to be clarified.

First of all, we would like to thank the Reviewer for the valuable comments, which helped to improve our work. We understand the Reviewer's concerns regarding the implementation of the multiple imputations in this study. We hope that the following response was able to clarify the source of the exposure data, why and how we implemented multilevel time raster multiple imputation to obtain the BMI trajectories, as well as why we believe that our imputation approach is valid and that this study adds valuable insights to the current literature.

The source of the exposure data was BMI measurements assessed and registered in the primary care records from the SIDIAP database by healthcare professionals. Although SIDIAP systematically collects BMI data since 2006, BMI information prior to this date is also available due to professionals recording data retrospectively and data transferred from paper to electronic health records in certain centers during the computerization process of the database. However, since the registration of BMI measurements prior to 2006 was not done homogeneously across Catalonia, we did not consider these measurements for this study. We clarified this in the methods sections (page 10, paragraph 3) of the manuscript:

- *To calculate BMI trajectories we extracted data on valid BMI measurements (before applying multiple imputations) assessed from January 1st, 2006 until December 31st, 2018.*

To address why and how we implemented multilevel time raster multiple imputation to obtain the BMI trajectories, we provided a more detailed explanation of this approach in the methods section of the manuscript (page 11; paragraph 4; page 12; paragraphs 1-3):

- *Calculation of BMI trajectories*
We applied multilevel time raster multiple imputation to BMI to obtain the BMI trajectories.¹⁹ This approach allows to impute irregularly spaced longitudinal data (such as unbalanced BMI assessments in an electronic health records database) relying on within- and between-patient information.
The "multilevel" component of this approach refers to the imputation model which was a linear mixed model. The cluster variable of this model was each individual's identifier. Level 1 variables (ie, that can vary within individuals) were the age at the BMI measurement and indicator variables of cancer, cardiometabolic conditions (ie, hypertension, type 2 diabetes, and cardiovascular diseases), and bariatric surgery, for which the value was 1 if the individual

had been diagnosed with the condition/disease or had the procedure before the BMI assessment (for cancer, 1 year prior), 0 if otherwise (as the diagnosis of these conditions and these procedures can lead to changes in BMI). Level 2 variables (ie, that vary between individuals) were sex, the MEDEA deprivation index, smoking status, alcohol intake, geographic region of nationality, the Charlson Comorbidity index (an indicator of multimorbidity composed of 19 items), diagnosis of different cancer types, and follow-up time.

The “time raster” component of this approach was used to homogenize the times of BMI measurement. We imputed BMI at six not-equally-spaced age points (18, 30, 40, 55, 70, and 90+ years of age). We used a B-spline of degree 1 to discretize the time of BMI measurement: considering all valid available BMI measurements in an individual’s health record, we attributed weights to the spline according to whether the age at measurement of the real BMI measurement coincided with one of the age points of interest. More details on this approach can be consulted in Appendix 1.

To construct the life-course trajectories, we joined two contiguous BMI measurements (ie, between two consecutive age points) with a straight line. This method has previously been used to assess longitudinal changes in BMI in SIDIAP and elsewhere.^{19,20} To implement the multilevel time raster multiple imputations we used the library MICE 3.13.0 available for the software R version 4.0.3.

With this approach, we were not only imputing data prior to 2006 but also data after 2006. This methodology uses information available from all individuals aged between 18 to 90+ years (n=5,279,567; population of the first step of the statistical analysis) in SIDIAP to impute BMI at several age points. At this step, we did not include the date of the BMI measurement as a variable in the linear mixed model nor the “time raster” to obtain the imputed BMIs, but the age of the BMI measurement instead. Furthermore, the applied methodology estimates (it does not predict) the BMI of the participants based on the likelihood with other BMI measurements of the participant whose BMI assessments are being imputed as well as other participants’ measurements “*matched*” on several characteristics (sex, the MEDEA deprivation index, smoking status, alcohol intake, geographic region of nationality, the Charlson Comorbidity index, diagnosis of different cancer types, and follow-up time). For each individual, this method produces five (n of imputations) different estimations per time point to account for the uncertainty surrounding each estimation.

In the future, we hope to be able to replicate this study with longer observed follow-up periods. However, as of today, we believe that our study still adds new and relevant information to the literature about the contribution of the duration, degree, and age of onset of overweight and obesity to the risk of cancer incidence, despite its limitations. Firstly, in Figure 1 of Appendix 1 (Supplementary Material) we showed that the BMI distribution between the observed and imputed datasets was very similar. Secondly, to our knowledge, prior studies of the field did not rely either on observed and directly measured BMI information for the whole period considered in their BMI trajectories (in such studies, BMI measurements capturing assessments many years prior to index date were self-reported). Despite the fact that the BMI trajectories of the present study were obtained using a multilevel time raster multiple imputation to BMI approach, these trajectories were based on objectively collected measures of BMI, which constitutes a different approach in comparison with previous studies. We were reassured regarding the quality of our exposures because the associations between BMI-derived longitudinal exposures and risk of specific cancer types that have been previously studied (colorectal, breast postmenopausal, endometrium, kidney, and multiple myeloma cancers) were similar to ours, even though those studies used other sources of BMI measurements.¹⁻⁴ On top of replicating previous

findings, our study contributes to the current knowledge by formally comparing cancer risk estimates of longitudinal exposures to those of baseline BMI and by also analyzing less common malignancies.

We expanded the limitations section to be clearer about the scope and disadvantages of the multiple imputations approach that we implemented (page 8, paragraph 2):

- *Our findings should be interpreted in light of some limitations. The major limitation is that due to the length of follow-up available (12 years), we exclusively relied on multiple imputed BMI measurements for the exposure window. It is important to clarify that the applied methodology aimed to estimate (not to predict) the BMI of the participants based on the likelihood with other BMI measurements of the participant whose BMI assessments were being imputed as well as other participants' measurements "matched" on several characteristics. To account for the uncertainty surrounding each BMI estimation, we had 5 imputed BMI trajectories per participant. Nevertheless, the implementation of this approach could have introduced exposure misclassification bias. We aimed to reduce this possibility by using high-quality BMI measurements (measured by health professionals and with a distribution shown to be similar to representative studies of the Spanish population)^{13,40} and by including data on all adults in SIDIAP (n=5,279,567) for the multiple imputations. We were also empirically reassured about the quality of our exposures given that we found similar associations between BMI-derived longitudinal exposures and risk of specific cancer types that have been previously studied (colorectal, breast postmenopausal, endometrium, kidney, and multiple myeloma cancers).^{2,7-10}*

References

1. Arnold M, Freisling H, Stolzenberg-Solomon R, et al. Overweight duration in older adults and cancer risk: a study of cohorts in Europe and the United States. *European Journal of Epidemiology*. 2016;31(9):893-904. doi:10.1007/s10654-016-0169-z
2. Noh H, Charvat H, Freisling H, et al. Cumulative exposure to premenopausal obesity and risk of postmenopausal cancer: A population-based study in Icelandic women. *International Journal of Cancer*. 2020;147(3):793-802. doi:https://doi.org/10.1002/ijc.32805
3. Arnold M, Jiang L, Stefanick ML, et al. Duration of Adulthood Overweight, Obesity, and Cancer Risk in the Women's Health Initiative: A Longitudinal Study from the United States. *PLOS Medicine*. 2016;13(8):e1002081-. https://doi.org/10.1371/journal.pmed.1002081
4. Marinac CR, Birmann BM, Lee IM, et al. Body mass index throughout adulthood, physical activity, and risk of multiple myeloma: a prospective analysis in three large cohorts. *British Journal of Cancer*. 2018;118(7):1013-1019. doi:10.1038/s41416-018-0010-4

On page 5, lines 139-147. It is not clear if the potential confounders considered were time varying or captured only at baseline or last follow-up.

We agree with the Reviewer that this was not clear in the text. Potential confounders were assessed only at one point in time. Age was defined at index date (January 1st, 2009). The MEDEA deprivation index was defined on December 31st, 2018 (date of data extraction), due to data availability. Smoking status and alcohol intake were defined as the closest registry to the index date. Sex and geographic region of nationality are available for all the individuals registered in the SIDIAP database and are assumed to remain unchanged over time by the database setup. We have added these details to the manuscript (page 11, paragraph 2):

- *The MEDEA deprivation index was defined based on the census tract where the participants were residing on December 31st, 2018 (date of data extraction). For smoking status and alcohol intake, we selected the closest registry to the index date.*

Reviewer #3 (Remarks to the Author):

The manuscript under consideration evaluates the relationship between a number of measures of adiposity (including baseline BMI and six longitudinal BMI-derived exposures for overweight and obesity [duration of, cumulative exposure to, and age of onset of BMI \geq 25 and \geq 30kg/m², respectively]) and the incidence of 26 different cancer types in a population-based cohort of 2.6 million adults in Spain. BMI measures were abstracted from medical records. Multiple imputation was used to calculate weight trajectories between ages 18 and 40. The authors identified a relationship between both baseline BMI and longitudinal measures of adiposity in 10 malignancies (uterine, kidney, gallbladder, thyroid, breast (postmenopausal), brain, leukemia, multiple myeloma, colorectal, and liver and, among never smokers, with head and neck and bladder cancers. Longitudinal exposures, but not baseline BMI, were positively associated with the risk of a number of additional malignancies (non-Hodgkin lymphoma, melanoma, and ovary, prostate, pancreas, and stomach cancers).

Although many prior publications have focused on the relationship between measures of adiposity (including longitudinal measures of adiposity) and cancer risk, the novelty of this manuscript lies in the fact that it reports on relationships between cancer risk and objectively collected measures of BMI and that the data set is large enough to provide sufficient power to evaluate the relationship between adiposity and less common malignancies. The authors also conducted a number of sensitivity analyses to further explore their findings. The manuscript is well written, although the results are a bit dense and might benefit from a table in addition to the figures presented to better depict the many relationships reported.

We thank the Reviewer for reviewing our manuscript. The suggestions made were very helpful and important to improve the quality of the manuscript.

We agree with the reviewer that the results can be seen as dense and that a table could be useful to better depict the different reported associations. Therefore, we added a supplementary table (Figure S15) summarizing the main findings of this study and we referenced this Figure in the discussion of the manuscript (page 6, paragraph 1):

- *(A table summarizing the main findings of this study can be consulted in Figure S15).*

This figure has also been mentioned in other parts of the discussion (page 6, paragraphs 2, 3, 4; page 7, paragraph 2).

My major concern is that the authors have created two reference categories of adiposity: BMI 25kg/m² and BMI 30kg/m². Many other publications have demonstrated a relationship between BMI 30kg/m², but there are fewer data suggesting that a BMI in the 25-30kg/m² range is associated with an increased risk of cancer. Given the size of the data set, it would be helpful to better characterize whether patients with a BMI in the overweight range are at increased risk of malignancy, vs this excess risk being driven by increased risk in individuals with a BMI in the obese range.

We thank the Reviewer for raising this topic. In the initial version of the manuscript, we decided to categorize the BMI-derived exposures into \geq 25 kg/m² and \geq 30 kg/m² to be consistent with the previous studies using these exposures (1-7). However, we agree that the analysis proposed by the Reviewer would add relevant information to the study which would contribute with further novel findings to the

current literature. We have added a secondary analysis in which we restricted the three longitudinal exposures to a BMI ≥ 25 and < 30 kg/m². We have modified the methods (page 13, paragraph 3) section and appendix 2 (where we described the results of the secondary and sensitivity analyses) accordingly:

- *We conducted five secondary analyses to contextualize our findings. [...] We recalculated the exposures restricting them to BMIs ≥ 25 and < 30 kg/m² to investigate the independent effect of overweight (from obesity) in relation to cancer risk.*
- *The associations with the recalculated exposures (restricted to BMIs ≥ 25 and < 30 kg/m² to investigate the independent effect of overweight from obesity in relation to cancer risk) were more attenuated than those of the main analyses (where we analyzed any BMIs ≥ 25) (Figure S12). Also, several statistically-significant associations in the main analysis became null in this secondary one (8 out of 14 for duration of, 5 out of 13 for cumulative exposure to, and 3 out of 13 for age of onset of a BMI ≥ 25 and < 30 kg/m²). However, the point estimates of the associations that became null were all ≥ 1 (eg, HR of gallbladder and biliary tract for duration of BMI ≥ 25 in the main analysis: 1.11 [1.05-1.18] vs for duration of BMI ≥ 25 and < 30 in the secondary analysis 1.03 [0.98-1.09]) (Figure S12).*

We also added Figure S12 to the supplementary material which contains the results of this analysis.

References

1. Stolzenberg-Solomon RZ, Schairer C, Moore S, Hollenbeck A, Silverman DT. Lifetime adiposity and risk of pancreatic cancer in the NIH-AARP Diet and Health Study cohort. *The American Journal of Clinical Nutrition*. 2013;98(4):1057-1065. doi:10.3945/ajcn.113.058123
2. Arnold M, Freisling H, Stolzenberg-Solomon R, et al. Overweight duration in older adults and cancer risk: a study of cohorts in Europe and the United States. *European Journal of Epidemiology*. 2016;31(9):893-904. doi:10.1007/s10654-016-0169-z
3. Noh H, Charvat H, Freisling H, et al. Cumulative exposure to premenopausal obesity and risk of postmenopausal cancer: A population-based study in Icelandic women. *International Journal of Cancer*. 2020;147(3):793-802. doi:<https://doi.org/10.1002/ijc.32805>
4. Arnold M, Jiang L, Stefanick ML, et al. Duration of Adulthood Overweight, Obesity, and Cancer Risk in the Women's Health Initiative: A Longitudinal Study from the United States. *PLOS Medicine*. 2016;13(8):e1002081-. <https://doi.org/10.1371/journal.pmed.1002081>
5. Marinac CR, Birmann BM, Lee IM, et al. Body mass index throughout adulthood, physical activity, and risk of multiple myeloma: a prospective analysis in three large cohorts. *British Journal of Cancer*. 2018;118(7):1013-1019. doi:10.1038/s41416-018-0010-4
6. Abdullah A, Wolfe R, Stoelwinder JU, et al. The number of years lived with obesity and the risk of all-cause and cause-specific mortality. *International Journal of Epidemiology*. 2011;40(4):985-996. doi:10.1093/ije/dyr018
7. Abdullah A, Amin FA, Stoelwinder J, et al. Estimating the risk of cardiovascular disease using an obese-years metric. *BMJ Open*. 2014;4(9):e005629. doi:10.1136/bmjopen-2014-005629

How was pregnancy accounted for in the weight trajectories? Given that more than 50% of the individuals in this cohort were women of childbearing years, pregnancy weights could confound these analyses given the use of multiple imputation to calculate missing weights (especially since women may be more likely to seek medical care and thus have recorded weights during pregnancy).

We agree with the Reviewer that accounting for pregnancy is important to correctly depict the BMI trajectories of women in childbearing years. While calculating the BMI trajectories, we did not consider BMI measurements assessed during pregnancy (from the 3rd month of pregnancy until 2 months after delivery). In the first version of the manuscript, we described this in the supplementary material (Appendix 1), but in the current version of the manuscript we have added this to the main text as well (page 10, paragraph 3):

- *To be considered valid BMIs, the BMI measurements had to be i) comprised between 15kg/m² and 60kg/m² (ie, extremely low values could be indicative of an underlying disease, and extremely low/high values could be due to data entry errors in medical records); ii) measured at or after 18 years; iii) not measured during pregnancy (from the 3rd month of pregnancy until 2 months after delivery).*

It would also be helpful to understand how many weights were available for individuals included in these analyses. The supplemental materials include a detailed description of the imputation methods used to calculate missing weights, but there is no information about the average number of weight measurements available. It is helpful that sensitivity analyses omitting individuals with just one weight did not show significantly different outcomes, but it would be helpful to understand how much of the data reported were actual measurements vs imputed values.

Following the Reviewer's valid comment, we added a supplementary table (Table S2) describing the baseline characteristics of the study population with and without a BMI assessment and we described this table in the results (page 3, paragraph 3) section of the manuscript:

- *Of the included participants, 2,081,840 (79%) had at least one BMI assessment in their electronic health records while 564,045 (21%) had none (Table S2). Individuals without a BMI assessment were more likely to be men and younger compared to those with a BMI assessment. The former also had a higher proportion of non-Spanish, individuals living in the least deprived areas of Catalonia and people with no comorbidities, compared to the latter (Table S2).*

Were any adjustments made for multiple testing? The authors report outcomes for 26 cancers using many different measures of weight at particular timepoints and over time. The risk of false positive results is particularly concerning given the narrowly significant associations in some malignancies and the finding in others that some adiposity measures were significantly associated with cancer risk while others were not (for example in melanoma and ovarian cancer). At the very least, this needs to be addressed in the limitations section of the manuscript.

In the first version of the manuscript that we submitted, we decided not to adjust for multiple testing because we had a pre-specified hypothesis (adiposity may be related to more cancer types than currently recognized in the adiposity-cancer literature). From our point of view, the present study differs from studies that are interested in statistically-significant associations between different unrelated exposures and numerous outcomes in the sense that our different exposures capture the same concept (adiposity). This being said, we understand the Reviewer's concern on this topic and for such reason, we have i) added a sensitivity analysis in which we applied the Bonferroni correction to counteract the multiple comparisons problem; ii) addressed this issue in the limitations section of the manuscript; and iii) tried to be wary in the interpretation of our findings (especially when only one exposure was linked to a specific cancer type).

The i)sensitivity analysis is mentioned in the methods and results sections (page 14, paragraph 2 and page 5, paragraph 2) of the manuscript as well as in appendix 2 (where we described the results of the secondary and sensitivity analyses):

- *We conducted four sensitivity analyses to assess the robustness of our findings. We [...] iv)applied the Bonferroni correction to counteract the fact that we are testing multiple comparisons. Our confidence intervals were corrected from 95% to 99% $[[1-(0.05/6)]x100]$ considering that we are analyzing six novel exposures (BMI at index date was included for comparability purposes) and cancer risk (each cancer type being considered as a different and specific disease).*
- *Overall, our results were similar to those from four sensitivity analyses although there were some differences in the sensitivity analysis in which we applied the Bonferroni correction (95% CIs changed to 99%) (eg, while no changes were seen for the duration of a BMI \geq 25 kg/m² exposure between the main and the sensitivity analysis, four out of ten positive associations became null for the age of onset of a BMI \geq 30 kg/m² exposure) (Figure S13 and S14).*
- *Further, when we applied the Bonferroni correction to counteract the fact that we were testing multiple comparisons, some of the positive associations of the main analysis became null (Figure S14). Longer duration of a BMI \geq 30 kg/m² was positively associated with the risk of 9 cancers in the sensitivity analysis instead of 12 in the main analysis (affecting non-Hodgkin lymphoma and cancers of the brain and the CNS and liver). Higher cumulative exposure to a BMI \geq 25 kg/m² with 12 instead of 13 cancers (no longer with bladder cancer). Higher cumulative exposure to a BMI \geq 30 kg/m² with 10 instead of 11 cancers (no longer with brain and the CNS cancer). Age of onset of a BMI \geq 25 kg/m² with 8 instead of 11 cancers (no longer with leukemia, and cancers of the liver and pancreas). Age of onset of a BMI \geq 30 kg/m² with 6 instead of 10 cancers (no longer with leukemia, multiple myeloma, and cancers of the stomach and gallbladder and biliary tract). Finally, BMI at index date with 8 instead of 10 cancers (no longer with cancers of the brain and the CNS and liver) (Figure 1, Figure S14).*

The findings of this analysis are presented in Figure S14 of the supplementary material.

Furthermore, we ii)expanded the limitations section of the manuscript to address this issue (page 9, paragraph 1):

- *Another weakness of this study is that we performed multiple tests which could have resulted in false positive associations. Although we were cautious in the interpretation of our main findings (ie, we emphasized consistent associations between the different exposures and a specific cancer type instead of focussing on p-values in isolation), to reduce the likelihood of false positive findings we also implemented the Bonferroni correction (sometimes criticized as being overly conservative) in a sensitivity analysis which revealed overall consistent findings (with some exceptions) with our main analysis (Figure S14).*

Finally, we iii)have tried to be cautious about the interpretation of the findings of this study and emphasized consistent associations between the different exposures and a specific cancer type instead of focussing on p-values in isolation. For instance, although we found positive associations between the different exposures and 18 cancer types, four cancer types were only linked to one exposure (malignant melanoma of the skin and cancers of the prostate, pancreas, and stomach) which was cautiously reasoned in the discussion section (page 7, paragraph 2):

- *Moreover, duration of BMI \geq 25 kg/m² was the only exposure positively associated with the risk of malignant melanoma of the skin and prostate cancers (Figure S15). This is in line with what*

was observed in the non-linear analysis of the association between BMI at index date and risk of these cancers (in this and other studies), where an inverted U-shaped association was found, indicating a higher risk of these cancers only for BMIs in the overweight range.^{40,41} Future research should focus on confirming these associations and on understanding the pathways by which only being overweight (and a longer duration of it) could have a harmful effect on the risk of these cancers. On the other hand, while higher levels of BMI have been convincingly associated with risk of pancreatic and gastric cardia cancers,² in our study we only found a positive association with respect to age of onset of a BMI \geq 25 (\geq 30) kg/m² (Figure S15). The lack of association with stomach cancer for other exposures could be due to our inability to distinguish gastric cardia (obesity-related) from non-cardia cancers (in Spain, the incidence of the non-obesity-related subsite of this cancer is higher than the obesity-related subsite) (Figure S15).⁴²

Reviewer comments, second round –

Reviewer #1 (Remarks to the Author):

The authors have addressed most of my concerns but there are still a few points that may need clarification:

1. How were Level 1 and Level 2 variables chosen for multilevel time raster multiple imputations?
2. The authors computed 5 BMI trajectories for each individual. Which of these trajectories did the author use to calculate longitudinal BMI exposures? Did the authors use the average of these trajectories or else?
3. What does "/" mean in Figure S15?

Reviewer #2 (Remarks to the Author):

The authors have adequately addressed my initial concerns.

Reviewer #3 (Remarks to the Author):

The authors have done a nice job in responding to the prior round of review. I have no further comments.

Manuscript ID: NCOMMS-22-14260-A

Title of the article: Longitudinal body mass index and cancer risk: a cohort study of 2.6 million Catalan adults

Dear Reviewer,

We are grateful to the Reviewer for the time and expertise invested in this review. We have answered the three comments and implemented the suggestions made in the manuscript when appropriate. A list of the numbered detailed responses is given below with the following format:

- **Black (bold): a copy of the text written by the Reviewer**
- *Blue (normal): our response*
- *Blue (italic): a copy of the modified text from the manuscript*

We hope our responses fully addressed the Reviewer's concerns and we wish to submit a revised version of the manuscript for further consideration in this journal.

Looking forward to the outcome of your assessment.

Yours sincerely,

Dr. Talita Duarte-Salles, Fundació Institut Universitari per a la recerca a l'Atenció Primària de Salut Jordi Gol i Gurina (IDIAPJGol), Barcelona, Spain

Dr. Heinz Freisling, International Agency for Research on Cancer-World Health Organization, Lyon, France

Dr. Veronica Davila-Batista, Endocrinology Department, Complejo Hospitalario Universitario Insular Materno Infantil, Las Palmas de GC, Canary Health Service, Las Palmas, Spain

The authors have addressed most of my concerns but there are still a few points that may need clarification:

1. How were Level 1 and Level 2 variables chosen for multilevel time raster multiple imputations?

We agree with the Reviewer that it is important to clarify how these variables were selected.

Level 1 variables are variables that are related to BMI values that vary within individuals (i.e., related to BMI values of a same person over time). In particular, we focused on indicator variables of diagnosis of cancer and cardiometabolic conditions (ie, hypertension, type 2 diabetes, and cardiovascular diseases) and bariatric surgery. These were chosen based on clinical knowledge (e.g., a diagnosis of type 2 diabetes can trigger changes in weight) as well as due to the quality of the variables' registry in the SIDIAP database (algorithms to capture these concepts have been developed in the database and comparisons with gold standards have been conducted).

Level 2 variables are variables that are related to the BMI values that vary between individuals. The Level 2 variables that we included in this study were based on the recommendations for variable selection in multiple imputations of Pedersen et al. (2017) and Moons et al. (2006) (1, 2). The authors recommend to include exposures, covariates, and time of follow up/outcomes (when using a time-to-event analysis) as well as auxiliary variables to impute variables with missing data. Therefore, we included all the variables of the main Cox proportional hazard (PH) models (exposure and covariates -which were selected based on a directed acyclic graph); the follow-up time and diagnosis of different cancer types (related to the outcome), and the Charlson-comorbidity index (as an auxiliary variable) to increase the precision and minimize the bias of the imputations (1,2).

We included this information in the methods section of the manuscript (paragraph 1, page 12):

- *Level 1 variables [...] were arbitrarily chosen based on clinical knowledge and due to the quality of the variables' registry in the SIDIAP database.*
- *Level 2 variables [...] were selected based on the multiple imputation literature stating that one should include exposures, covariates, and time of follow up/outcomes (when using a time-to-event analysis) as well as auxiliary variables to impute variables with missing data.^{34,35}*

References

- 1) Pedersen AB, Mikkelsen EM, Cronin-Fenton D, Kristensen NR, Pham TM, Pedersen L, Petersen I. Missing data and multiple imputation in clinical epidemiological research. *Clin Epidemiol.* 2017 Mar 15;9:157-166. doi: 10.2147/CLEP.S129785. PMID: 28352203; PMCID: PMC5358992.
- 2) Moons KG, Donders RA, Stijnen T, Harrell FE Jr. Using the outcome for imputation of missing predictor values was preferred. *J Clin Epidemiol.* 2006; 59(10):1092-1101.

2. The authors computed 5 BMI trajectories for each individual. Which of these trajectories did the author use to calculate longitudinal BMI exposures? Did the authors use the average of these trajectories or else?

We thank the Reviewer for raising this topic. In this study, we followed the usual pipeline for statistical analyses using imputed datasets. We started by imputing the five BMI trajectories for every individual as pointed out by the Reviewer. Secondly, we performed the statistical analyses (Cox PH models) in each of the five imputed datasets. Finally, we pooled the five results of the analyses to obtain one estimate (with its respective variance) using Rubin's rule (1). We did not average the results as this would have ignored the variability among imputations, leading to incorrect standard errors, confidence intervals, and p-values (2). In the present study, we performed both the multilevel time raster multiple imputation and the respective pooling of results using the library "MICE" 3.13.0 available for the software R, version 4.0.3.

To address the Reviewer's valid point, we expanded the methods section (page 11, paragraph 4; page 13, paragraph 2) of the manuscript and the supplementary material (Appendix 2, page 33, paragraph 1):

- *We applied multilevel time raster multiple imputation to BMI to obtain the BMI trajectories (five imputed trajectories per individual).³⁴*
- *We investigated the association between each of the exposures with the risk of the 26 cancer types by running Cox proportional hazard models with age as the underlying time metric in each of the five imputed datasets and pooling the results using Rubin's rule.^{37,38}*
- *After imputing the five BMI trajectories for every individual, we used these trajectories to construct longitudinal BMI-derived exposures among the study participants. Then, we investigated the association between the exposures and cancer risk using Cox Proportional Hazards models. The models' estimates were pooled using Rubin's rule (3). To implement the multilevel time raster multiple imputations and the pooling of results we used the library MICE 3.13.0 available for the software R version 4.0.3.*

References

1. Rubin, D.B. *Multiple Imputation for Nonresponse in Surveys*. J. Wiley & Sons, New York; 1987.
2. van Buuren S. *Flexible Imputation of Missing Data: Time Raster Imputation*. 2nd Edition. Chapman & Hall/CRC; 2012.

3. What does "/" mean in Figure S15?

We thank the reviewer for this comment. We agree that this was not clear in the previous version of the manuscript. Cells marked with "/" in Figure S12 (in the previous version S15) represent non-linear associations according to the log-likelihood test of linearity that visually appear to be linear or do not exhibit a specific non-linear pattern.

We have expanded the notes of Figure S12 to facilitate the comprehension of the figure:

- *Notes: [...] Cells filled in red denote positive linear associations and green denote negative linear associations. Letters in the intersection between exposures and cancer types represent the shape of observed non-linear associations. Cells marked with "L" represent L-shaped associations between the exposures and specific cancer types. Cells marked with "J" represent J-shaped associations between the exposures and specific cancer types. Cells marked with "∩" represent inverted-U-shaped associations*

between the exposures and specific cancer types. Cells marked with “U” represent U-shaped associations between the exposures and specific cancer types. Cells marked with “/” represent non-linear associations according to the log-likelihood test of linearity that visually appear to be linear or do not exhibit a specific non-linear pattern. Cells marked with “” represent positive associations between the exposures and specific cancer types among never smokers. For examples, please refer to Figures 2, 3, 4, S2 and S7.*